# A simulation experiment-based assessment of retrievals of above-cloud temperature and water vapor using hyperspectral infrared sounder

Jing Feng[1], Yi Huang[1], and Zhipeng Qu[1,2]

[1]Department of Atmospheric and Oceanic Sciences, McGill University
[2]Observations-Based Research Section, Environment and Climate Change Canada

**Correspondence:** Jing Feng (jing.feng3@mail.mcgill.ca)

**Abstract.** Measuring atmospheric conditions above convective storms from space-borne instruments is challenging. The operational retrieval framework of current hyperspectral infrared sounders adopts a cloud-clearing scheme that is unreliable in overcast conditions. To overcome this issue, previous studies have developed an optimal estimation method that retrieves temperature and humidity above high, thick clouds, assuming a slab of cloud. In this study, we find that variations in the effective radius and density of cloud ice near the top of convective clouds lead to non-negligible spectral uncertainties in simulated infrared radiance spectra. These uncertainties cannot be fully eliminated by the slab-cloud assumption. To address this problem, a synergistic retrieval method is developed. This method retrieves temperature, water vapor, and cloud properties simultaneously by incorporating observations from active sensors in synergy with infrared radiance spectra. A simulation experiment is conducted to evaluate the performance of different retrieval strategies using synthetic radiance data of the Atmospheric InfraRed Sounder (AIRS) and cloud data of CloudSat/CALIPSO. In this simulation experiment, we simulate infrared radiance spectra from convective storms through the combination of a numerical weather prediction model and a radiative transfer model. The simulation experiment shows that the synergistic method is advantageous as it shows strong retrieval sensitivity to the temperature and IWC near the cloud top. The synergistic method reduces root-mean-square errors in temperature and column integrated water vapor by more than half compared to the prior knowledge based on the climatology. It can also improve the quantification of ice water content and effective radius compared to the prior knowledge based on the retrieval of the active sensors. Our results suggest that existing infrared hyperspectral sounders can detect the spatial distribution of temperature and humidity anomalies above convective storms.

## 1 Introduction

Water vapor in the upper-troposphere and lower-stratosphere (UTLS) plays an essential role in the Earth's climate system due to its important radiative (Huang et al., 2010; Dessler et al., 2013) and chemical effects (Shindell, 2001; Kirk-Davidoff et al., 1999; Anderson et al., 2012).

Our understanding of UTLS water vapor has long been informed by accurate in-situ observations carried out during aircraft and balloon campaigns. Long-term records provided by balloon-borne observations have suggested a decadal increase

in stratospheric water vapor (Oltmans et al., 2000; Rosenlof et al., 2001; Hurst et al., 2011) but a decadal cooling in tropical
tropopause temperature over the same period (Rosenlof et al., 2001; Randel et al., 2004). These contradictory trends between
water vapor and temperature are not well reproduced by reanalysis products (Davis et al., 2017) and the key processes at play
are still under debate. This increase in UTLS water vapor, if true, may accelerate the decadal rate of surface warming through
its impact on thermal radiation (Solomon et al., 2010). While balloon-borne instrument suggests possible changes in UTLS
water vapor, aircraft campaigns reveal that UTLS water vapor can be highly variable under the influence of deep convection.
By sampling plumes from convective detrainment, these campaigns have found that overshooting deep convection can increase
the UTLS water vapor by injecting moist plumes or ice particles that sublimate in a warmer environment (e.g., Corti et al.,
2008; Schiller et al., 2009; Anderson et al., 2012; Sun and Huang, 2015; Smith et al., 2017). Despite substantial evidence of
convective hydration, it has been argued that the overall impact of convection on the global UTLS water vapor budget might
be negligible (e.g., Ueyama et al., 2018; Schoeberl et al., 2019; Randel and Park, 2019).

Therefore, long-term, global observations of UTLS water vapor, especially above convective storms, are essential. However,
the operational global radiosonde network does not perform well in cold and low-pressure environments such as the UTLS
(Kley, 2000). Moreover, while satellite observational products have been extensively used to investigate the spatial and temporal variability of UTLS water vapor (Sun and Huang, 2015; Randel and Park, 2019; Yu et al., 2020; Wang and Jiang, 2019;
Jiang et al., 2020), these products have some limitations. Although limb-viewing and solar occultation instruments are sensitive to the UTLS region, they are not suitable for detecting the small-scale variability above convective storms because the
horizontal sampling footprints of these instruments are larger than 100 km. Furthermore, contamination by convective clouds
leads to higher uncertainty in the current product of microwave sounders (such as MLSv4.2, Livesey et al., 2017) due to strong
scattering. Moreover, limited by the occurrence of solar occultation, instruments using this technique do not provide sufficient
sampling to study convective events.

Meanwhile, the current hyperspectral sounding framework of NOAA and NASA adopts a cloud-clearing scheme (Susskind
et al., 2003; Gambacorta et al., 2014). This scheme infers the radiance of clear scenes from adjacent $3 \times 3$ instrument fields-of-view (FOVs) with varying cloud amounts, assuming the same temperature and atmospheric absorber (including water vapor)
fields in all FOV footprints ($\sim 13.5$ km). Consequently, such a cloud-clearing scheme would fail in overcast cloud conditions
(uniform cloud amounts in adjacent footprints) or when thermodynamic properties vary drastically among adjacent footprints.
For this reason, the current retrieval products from hyperspectral infrared sounders, including AIRS (the Atmospheric InfraRed
Sounder, Chahine et al., 2006), IASI (Infrared Atmospheric Sounding Interferometer, Blumstein et al., 2004), and CrIS (Cross-track Infrared Sounder, Bloom, 2001), are not reliable above convective storms.

Recently, researchers have demonstrated the feasibility of performing single-footprint retrievals in cloudy-sky conditions
from AIRS using an optimal estimation (OE) scheme (DeSouza-Machado et al., 2018; Irion et al., 2018; Feng and Huang,
2018). Using the same instrument, such single-footprint retrievals improve the spatial resolution from 40.5 km in the cloud-clearing scheme to 13.5 km. In these studies, DeSouza-Machado et al. (2018) used the *a priori* cloud state from a numerical
weather prediction model (NWP) and then adjusted the cloud state to match the observed brightness temperature of an infrared
window channel. Irion et al. (2018) retrieved cloud optical depth, effective radius, and cloud-top temperature, using the *a*

*priori* from collocated MODIS (Moderate Resolution Imaging Spectroradiometer, Platnick et al., 2003) observations. While DeSouza-Machado et al. (2018) and Irion et al. (2018) discussed the implementation of all-sky, single-footprint OE scheme in general, Feng and Huang (2018) focused especially on optically thick cloud conditions, for which they conducted a comprehensive information content analysis. They showed that the existing hyperspectral infrared sounders contain a substantial amount of degrees-of-freedom for signal (DFS, a higher DFS indicates a higher vertical resolution) in UTLS temperature ($\sim$5) and water vapor ($\sim$1). They also found that the presence of a thick cloud in the upper-troposphere increases the DFS compared to clear-sky conditions. By validating the retrieval using in-situ observations carried by aircraft campaigns, Feng and Huang (2018) evidenced that it is possible to detect both hydration and dehydration anomalies in the UTLS from current infrared hyperspectral sounders. In the case of optically thick clouds, e.g., deep convective clouds, these studies (DeSouza-Machado et al., 2018; Irion et al., 2018; Feng and Huang, 2018) similarly represent the cloud as a slab (optically thick and uniform layer) of ice clouds with fixed microphysics properties, based on *a priori* cloud states inferred from brightness temperature of an infrared window channel, NWP, or coincident passive cloud instrument (e.g., MODIS). Retrieval methods following this cloud assumption are referred to as slab-cloud methods hereinafter.

However, neglecting variability in cloud mass and microphysical properties leads to uncertainty in the thermal emission of the cloud, which greatly contributes to observed radiances at the top-of-atmosphere (TOA). Yang et al. (2013) showed that the scattering and absorption properties of ice clouds across the infrared spectra are greatly impacted by the size and shape of ice particles. Furthermore, deep convective clouds are typically associated with large temperature perturbations near the cloud top and drastic temperature decreases with altitude (Biondi et al., 2012). Considering an anomalous temperature field, inferring cloud top position from the brightness temperature of an infrared window channel, as in previous studies, can lead to biases (Sherwood et al., 2004). When the temperature lapse rate is large, the vertical distribution of ice content can influence the cloud thermal emission. Therefore, it is necessary to assess, and constrain, the impacts of these factors on retrieval accuracy.

The cloud uncertainties can be reduced by combining collocated observations from active sensors onboard the same satellite constellation. The A-Train satellite constellation provides a unique collocation between an orbital hyperspectral infrared sounder (i.e., AIRS) and active remote-sensing instruments, including the cloud profiling radar aboard CloudSat (Stephens et al., 2002) and CALIOP (Cloud-Aerosol Lidar with Orthogonal Polarization) aboard CALIPSO (Cloud-Aerosol Lidar and Infrared Pathfinder Satellite Observation, Winker et al., 2003). Along the A-Train orbit track, these instruments passed over nearly the same locations within 2 minutes of each other (before the year 2015). The nearest lidar (90 m $\times$ 90 m) and CPR footprints (2.5 km $\times$ 1.4 km) were typically located around 5 km from the center of the AIRS footprints (13.5 km $\times$ 13.5 km), well within the AIRS FOVs. DARDAR-Cloud (Delanoë and Hogan, 2008, 2010) is a joint product that combines radar reflectivity measurements from CPR and lidar attenuated backscatter ratio from CALIOP to provide ice water content (IWC) and effective radius profiles at each CPR footprint. Compared to passive instruments, this joint product is more sensitive to the vertical ice distribution near the cloud top, which can be essential to the thermal emission of the cloud. Here we aim to develop an optimal estimation method to retrieve temperature, water vapor, ice water content, and effective radius simultaneously by incorporating active cloud remote sensing products and infrared hyperspectra, using the DARDAR-Cloud product and AIRS

L1B observations to construct an example. A retrieval method that incorporates such collocated cloud products is referred to as a synergistic method.

In this paper, we first quantify the uncertainty in infrared radiance spectra induced by cloud optical properties. The performance of retrieval strategies following the slab-cloud and synergistic methods is then evaluated following a simulation experiment, emulating an implementation based on the AIRS L1B and DARDAR-Cloud products. This simulation experiment simulates observational signals from realistic temperature, humidity, and cloud fields above a deep convective event simulated by an NWP model. Section 2 describes the main components of this simulation experiment. We then implement different

retrieval strategies, as formulated in Section 2.3, to retrieve from synthetic observations. The results are evaluated in Section 3 by comparing retrievals to the prescribed truth. The application of the improved synergistic retrieval scheme to existing instruments is discussed in Section 4.

## 2   Method

The simulation experiment in this study consists of the following components:

1. a cloud-resolving NWP model, which is used to provide the 'Truth' of atmospheric conditions during a tropical cyclone event and to construct the *a priori* and test sets, as described in Section 2.1;

     2. a radiative transfer model, which is used to generate synthetic observations with the AIRS instrument specifications and as the forward model in the retrieval, as described in Section 2.2;

     3. retrieval algorithms as explained in Section 2.3; and

4. comparisons between the retrieved quantities and the NWP-generated truth in Section 3.

A tropical cyclone event is simulated because it generates a vast convective cloud system that covers a large spatial domain for contrasting the above-storm temperature and humidity fields. In the framework of this simulation experiment, we neglect the complexity in instrument scan geometry by assuming a nadir instrument viewing angle, uniform atmospheric conditions within one footprint, and availability of coincident cloud products for every sample. In reality, the scanning angle of AIRS

footprints that have the nearest CloudSat footprints within 6.5 km from the center is around 16 degrees off the nadir.

### 2.1   Numerical weather prediction model

In this study, we use the Global Environmental Multiscale model (GEM) of Environment and Climate Change Canada (hereafter ECCC, Côté et al., 1998; Girard et al., 2014) to provide a detailed and realistic representation of storm-impacted atmospheric and cloud profiles, following the study of Qu et al. (2020). The GEM model is formulated with non-hydrostatic

primitive equations with a terrain-following hybrid vertical grid. It can be run as a global model or a limited-area model and is capable of one-way self-nesting. For the experiments conducted here, three self-nested domains are used with areas of $3300\times3300$, $2000\times2000$, and $1024\times1024$ km$^2$ and horizontal grid-spacings of 10, 2.5, and 1 km, respectively, centered at

141°E, 16°N. All simulations use 67 vertical levels, with vertical grid-spacing $\Delta z \sim 250$ m in the UTLS region and a model top at 13.5 hPa (29.1 km). The simulation is initialized with conditions from the ECCC global atmospheric analysis at 00:00 UTC 16 May 2015. It runs for 24 hours until 00:00 UTC on 17 May 2015. The model spin-up time of 6 hours is used to assure the correct formation of clouds. Model outputs at 1 km horizontal grid-spacing are saved every 10 minutes. The subdomains of the 1 km simulation near the cyclone are used in the simulation experiment.

For the two high-resolution simulations with 2.5-km and 1-km horizontal grid-spacing, the double-moment version of the bulk cloud microphysics scheme of Milbrandt and Yau (2010a, b; hereinafter referred to as MY2) is used. This scheme predicts the mass mixing ratio for each of six hydrometeors including non-precipitating liquid droplets, ice crystals, rain, snow, graupel, and hail. Condensation (ice nucleation) is formed only upon reaching grid-scale supersaturation with respect to liquid (ice). In addition to the MY2 scheme, the planetary boundary-layer scheme (Bélair et al., 2005) and the shallow convection scheme (Bélair et al., 2005) can also produce cumulus, stratocumulus, and other low-level clouds, which are of less relevance to our UTLS-centric simulation experiment.

A snapshot from the 1-km resolution GEM simulation, 410 minutes after the initial time step, is used for the radiance simulation because a mature storm at this time has generated abundant convective clouds, which our retrieval approach targets. Figure 1 shows the atmospheric conditions at this time step, including the distributions of temperature and water vapor at 81 hPa, at which level the variance is largest. To mimic the satellite infrared image, we show the distribution of the brightness temperature in a window channel at 1231 cm$^{-1}$ (8.1 $\mu$m, $BT_{1231}$). A cold $BT_{1231}$ suggests a deep convective cloud (DCC) that extends to the tropopause level. Overshooting DCCs are often identified from satellite infrared images based on a warmer BT in a water vapor channel ($BT_{1419}$ cm$^{-1}$) relative to $BT_{1231}$, which can be attributed to water vapor emission above the cold point (Aumann and Ruzmaikin, 2013). The BT-based criterion is used to select retrieval samples, mimicking the scenario of using satellite infrared radiance measurements alone to identify overshooting DCCs, as performed in Feng and Huang (2018). Using the BT-based criterion, 9941 retrieval samples are identified, with their locations marked in Fig. 1. These samples are confirmed to be continuous, precipitating clouds that fully cover vertical ranges from near-ground to 380 K potential temperature. Among these samples, 100 profiles are randomly selected to construct a test set. The size of the samples is verified to meet the convergence requirement of the statistical evaluation conducted in Section 3. The rest of the simulated profiles, regardless of cloud conditions, numbering O($10^6$), are used to construct an *a priori* dataset to define the prior knowledge used in the retrieval in Section 2.3.

## 2.2 Radiative transfer model

This study uses the MODerate spectral resolution TRANsmittance, version 6.0 (MODTRAN 6.0) (Berk et al., 2014) to simulate infrared radiance spectra observed by satellite. MODTRAN 6.0 provides a line-by-line (LBL) algorithm that performs monochromatic calculations at the center of 0.001 cm$^{-1}$ sub-bins. Within each 0.2 cm$^{-1}$ spectral region, this method explicitly sums contributions from line centers while precomputing contributions from line tails. This algorithm has been validated against a benchmark radiation model, LBLRTM, showing less than 0.005 differences in atmospheric transmittance through most of the spectrum (Berk and Hawes, 2017). MODTRAN 6.0 accounts for both absorptive and scattering media in the atmo-

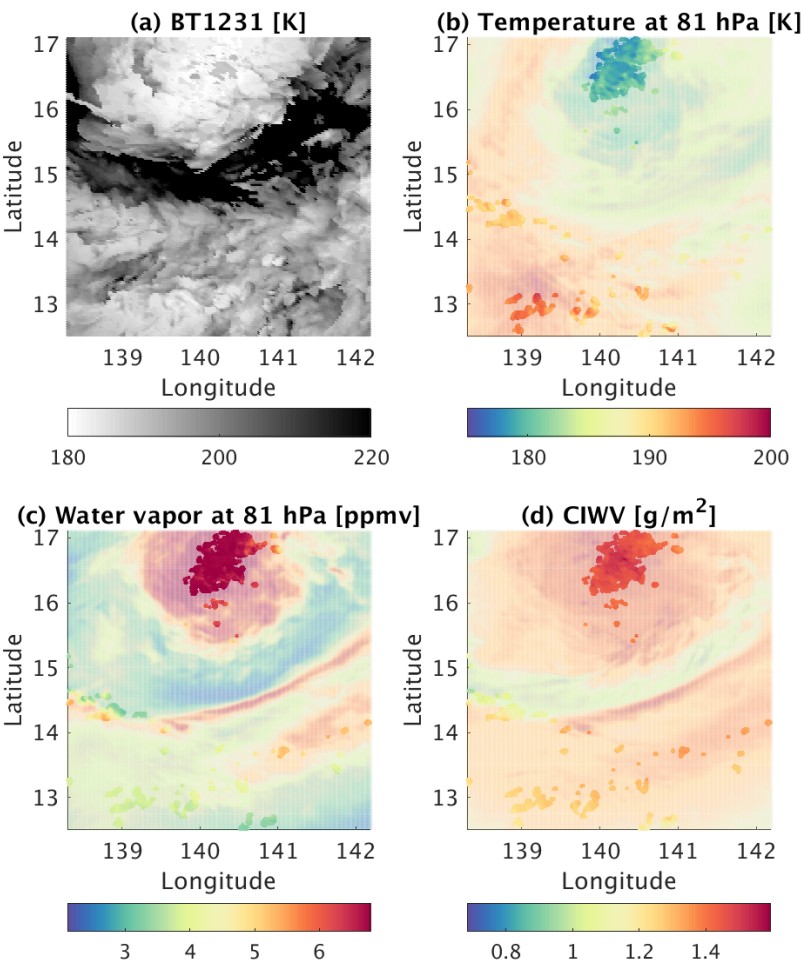

**Figure 1.** GEM-simulated atmospheric conditions used as the "Truth" in the simulation experiment. (a) Brightness temperature [K] at 1231 cm$^{-1}$. (b) Temperature [K] at 81 hPa. (c) Water vapor volume mixing ratio [ppmv] at 81 hPa. (d) Column integrated water vapor (CIWV) from 110 to 70 hPa. Solid color-coded dots mark the overshooting deep convective clouds sampled via BT-based criterion from which the test set is sampled to conduct the retrievals. Partially transparent colors show the rest of the simulated fields. The variable fields are taken at 410 minutes after the initial time step.

sphere by implementing a spherical refractive geometry package and the DISORT discrete ordinate model to solve the radiative transfer equation (Berk and Hawes, 2017).

In this study, we use MODTRAN 6.0 to simulate the all-sky radiances with user-defined atmospheric profiles. 80 fixed atmospheric pressure levels are used. Temperature, water vapor, and ice cloud (IWC) profiles from GEM simulations at 67 layers are input into the model. Above the GEM model top (13.5 hPa), the values from a standard tropical profile (McClatchey, 1972) are placed between 13.5 and 0.1 hPa. Other trace gases are fixed at a tropical mean value.

User-defined cloud extinction coefficients, single scatter albedo, and asymmetry factor (defined per unit mass of cloud ice) are added to the model, based on the cloud optical library of Yang et al. (2013). This cloud optical library provides a look-up table for the scattering, absorption, and polarization properties of ice particles of different habit shapes, roughness, and sizes. We parametrize the particle size distribution following microphysical data obtained from in-situ observations at temperature lower than -60 $^oC$ (Heymsfield et al., 2013; Baum et al., 2014). Following Baum et al. (2011) Appendix A-B, the mean extinction coefficients, mean single-scattering albedo, and mean asymmetry factor of the parametrized particle size distribution are obtained at individual wavelengths, for effective radii that range from 1 to 100 $\mu m$. These optical properties are then supplied to the radiative transfer calculations for specified effective radii and crystal habit mixtures. The optical depth of ice clouds in the DCC samples exceeds 100, which completely attenuates emission from liquid clouds. Liquid clouds are therefore neglected.

The instrument specifications of AIRS are used in the retrieval framework of this simulation experiment. This instrument has 2378 channels from 650 to 2665 $\mathrm{cm}^{-1}$. The radiometric noise of this instrument is obtained from the AIRS L1B product, which corresponds to a noise equivalent temperature difference (NEdT) around 0.3 K (at 250 K). This NEdT increases to around 0.5 K at 200 K reference temperature. Based on the radiometric quality of each channel, 1109 channels are selected. This rigorous channel selection also excludes O3 absorption channels (980-1140 $\mathrm{cm}^{-1}$), CH4 absorption channels (1255-1355 $\mathrm{cm}^{-1}$), and shortwave infrared channels (2400-2800 $\mathrm{cm}^{-1}$). Adopting the AIRS spectral response function, synthetic radiances are generated using MODTRAN with temperature, water vapor, and ice water content profiles from the test set described in Section 2.1. Effective radius profiles of the test set are prescribed according to the DARDAR-Cloud observations described in Section 2.2.1. A crystal habit mixture model (Baum et al., 2011) for tropical deep convective clouds are used for generating synthetic radiance spectra. Spectrally uncorrelated noise is generated and added to the synthetic radiance spectra. The noise at each channel follows the Gaussian distribution with its mean equal to the radiometric noise of the AIRS instrument. These infrared radiance spectra are used as synthetic observations in the simulation experiment.

## 2.2.1 Cloud induced uncertainties

Ice cloud impacts infrared radiance spectra via its thermal emission. Besides its temperature, the cloud thermal emission is influenced by the mass density of cloud ice and its optical properties, which include extinction coefficient, single-scattering albedo, and asymmetry factor. These optical properties are jointly affected by the particle size distribution, effective radius, habit, and surface roughness of ice particles, and are defined per unit mass in this study. In this section, we are interested in

whether the mass density of cloud ice and optical properties leave significant impacts on infrared radiance spectra. We also evaluate uncertainties of the forward model in simulating infrared radiance spectra with simplified cloud inputs.

The cloud-induced uncertainties in infrared radiance spectra are evaluated with regard to three factors: 1) variation in IWC, 2) variation in cloud optical properties caused by column to column (horizontal) variation in effective radius, and 3) variation in cloud optical properties caused by variation in crystal habit mixture and layer-to-layer (vertical) variation in effective radius. Uncertainties due to particle size distribution are not evaluated because of the lack of observation in its variability and its smaller impact compared to the other cloud variables considered here. The surface roughness of ice particles is neglected because it mainly affects the scattering angle (Yang et al., 2013), which plays a minor role in the infrared channels. To gain knowledge of cloud ice particles and their impacts on the infrared radiance spectrum and also to prescribe relevant information in the UTLS retrieval (see Section 2.3), we use the DARDAR-Cloud product to form a dataset of observations close to tropical cyclones, due to their relevance to the simulation experiment. The moment when A-Train satellites pass over tropical cyclones is identified by the CloudSat 2D-TC product (Tourville et al., 2015) for the year 2006 to 2016. Only overpasses in the western part of the Pacific are used. From these overpasses, we select DARDAR footprints that are within $1000 \, \mathrm{km}$ of the cyclone center locations. Based on the CloudSat-CLDCLASS product, 98293 of these footprints contain OT-DCCs that penetrate beyond 16 km in altitude. Each profile consists of IWC ($IWC$) and effective radius ($Re$) at a vertical resolution of 60 m.

Using the identified OT-DCC profiles from DARDAR-Cloud, we calculate the probability distribution function (PDF) of effective radii of ice particles at the topmost cloud layer. Figure 2 (a) shows that the ice particles are typically small, with an average effective radius of 21.5 $\mu$m and the 1st and 99th percentiles of 13.3 and 39.7 $\mu$m respectively. Using the same OT-DCC profiles, the mean and standard deviation (STD) of IWC profiles are shown in Fig.2 (b). The statistical calculations performed here exclude zero values. The average cloud top height is 16.7 km.

For tropical deep convective clouds, Baum et al. (2011) developed a habit mixture model as a function of ice particle sizes. Using this model, ice cloud optical properties are generated following the description in Section 2.2. A radiance spectrum calculated using this model is denoted with the subscript $'mix'$. Based on the habit mixture model, over $80\%$ of small ice particles in tropical deep convection are solid columns. Therefore, we also generate radiance from ice cloud optical properties using solid columns alone, which are denoted with the subscript $'sc'$.

100 profiles are selected from OT-DCC samples. For each sample, we calculate the upwelling infrared radiance, $R_{mix}(Re, IWC)$, using the IWC profile ($IWC$), effective radius profile ($Re$), and the habit mixture model developed by Baum et al. (2011). The mean temperature ($t_0$) and water vapor ($q_0$) profiles of the NWP simulation domain (Fig. 1) are used in the radiative transfer calculations.

Considering that the infrared radiance spectra may not be sensitive to vertical variations in cloud optical properties, we assume constant optical properties in all vertical layers of an atmospheric column and crystal habit of solid columns, to simplify the input cloud variables to MODTRAN. Following this assumption, we calculate $R_{sc}(Re_{opt}, IWC)$ using solid column alone and one effective radius value, $Re_{opt}$, for all vertical layers of an individual profile. This $Re_{opt}$ is solved iteratively and it minimizes the brightness temperature difference between $R_{mix}(Re, IWC)$ and $R_{sc}(Re_{opt}, IWC)$. The PDF of $Re_{opt}$ is shown by red in Fig. 2 (a), with an average of 34 $\mu$m ($Re_0$) and a STD of 11 $\mu$m. In practice, one may estimate the $Re_{opt}$

from the effective radius of a cloud layer where the optical depth measured from the cloud top reaches unity, with a root-mean-square-error (RMSE) of 1.6 $\mu m$ ($\sim 5\%$). RMSE spectra in $R_{sc}(Re_{opt}, IWC)$ relative to $R_{mix}(Re, IWC)$ is shown by the red solid curve in Fig.3 (a). The magnitude of the RMSE spectrum in the mid-infrared is around 0.1 K, confirming that the mid-infrared spectra are not sensitive to layer-to-layer variations in effective radius or mixtures of crystal habits differing from the solid column. Using constant cloud optical properties for the entire column of a tropical deep convective cloud can reasonably represent the mid-infrared emission spectra of the cloud. At wavenumbers higher than 1800 $cm^{-1}$, however, neglecting such variations in effective radius and crystal habits induces significant RMSE, as shown in Fig. 3 (a). The RMSE spectrum is also computed adopting an AIRS-like spectral response function, denoted as $\varepsilon_{synergistic}$, to represent the forward model uncertainty in the synergistic retrieval method introduced in Section 2.3.

In the following contents of this paper, $Re_{opt}$ determined for each profile as described above is used to represent the vertically constant effective radius value for characterizing cloud optical properties of a cloud column. It is also used to evaluate the spectral differences caused by IWC and column-to-column variations in cloud optical properties. We calculate infrared radiance spectra with mean effective radius ($Re_0$, 34 $\mu m$) or IWC profile ($IWC_0$), denoted as $R_{sc}(Re_0, IWC)$ and $R_{sc}(Re_{opt}, IWC_0)$, respectively. Then, perturbations of infrared spectra caused by variations in effective radius ($Re_{opt}$) are evaluated by the mean (blue curve) and the STD (grey shaded area) of the equivalent brightness temperature of $R_{sc}(Re_{opt}, IWC_0)$, as shown in Fig. 3 (b). Using a mean effective radius leads to a RMSE spectrum in $R_{sc}(Re_0, IWC)$ relative to $R_{sc}(Re_{opt}, IWC)$, which is shown by a red curve in Fig. 3 (b). Similar results are shown in Fig. 3 (c) for IWC.

In Fig.3 (b,c), the mean spectrum of OT-DCCs shows cold and relatively uniform brightness temperatures in the window and weak absorption channels that largely correspond to the emission from the cloud top. While variations in effective radii ($Re_{opt}$) and IWC have a weak effect on the strong absorption channels, they greatly impact the cloud emission, thus leading to large radiance variations in the window and weak absorption channels. As a result, the two RMSE spectra are similar. The RMSE due to column-to-column variations in effective radius ($Re_{opt}$) is around 1 K and the RMSE due to a varying IWC profile is around 3 K.

The RMSE spectra are further normalized with respect to the spectral mean, as shown in Fig.3 (d), to examine whether spectral signatures of effective radius and IWC are distinguishable from each other. Despite the overall similarity, effective radius affects the spectrally dependent extinction coefficients, leading to a tilted pattern across the infrared spectra, while the RMSE due to IWC is relatively uniform across the infrared window. Therefore, it is possible to distinguish the radiative signals of effective radius from those of IWC with a mid-infrared coverage characteristic of existing instruments. Interestingly, differences in the two normalized RMSE spectra are more prominent at lower wavenumbers ($\sim 200$ $cm^{-1}$), suggesting that far-infrared channels, e.g., from future instruments, such as FORUM (Palchetti et al., 2020) and TICFIRE (Blanchet et al., 2011), may be advantageous for the UTLS retrieval, which is beyond the scope of this simulation experiment but warrants future investigation.

For a comparison, we follow Feng and Huang (2018) to obtain the infrared spectra using the slab-cloud method. For each $R_{mix}(Re, IWC)$, we calculate the brightness temperature of a window channel at 1231 $cm^{-1}$. The idea of slab-cloud method used by Feng and Huang (2018) is to minimize the infrared radiance residual at this window channel by placing a slab of cloud

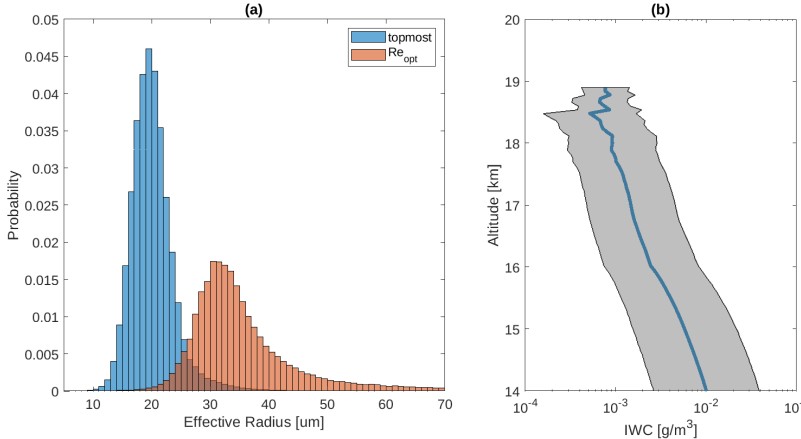

**Figure 2.** Cloud statistics based on 98293 overshooting deep convective samples from the DARDAR-Cloud dataset. The samples are selected within 1000 km of tropical cyclone center. (a) Histogram of the effective radius ($\mu$m) of cloud ice particles at the topmost layer (blue) and the effective radius for representing vertically uniform optical properties ($Re_{opt}$, red). (b) Mean IWC (blue curve) and STD of IWC (grey area).

at the vertical layer where the atmospheric temperature differs the least from $BT_{1231}$. This 500-m thick slab-cloud has uniform IWC of 1.5 g/m$^3$ and an effective radius of 34 $\mu$m. The temperature of this vertical layer is adjusted to $BT_{1231}$. With this prescribed cloud layer, radiance spectra are calculated again for each profile, denoted as $R_{sc}(Re_0, slab)$. The $BT_{1231}$ values of $R_{mix}(Re, IWC)$ and $R_{sc}(Re_0, slab)$ are identical. Consequently, differences between $R_{mix}(Re, IWC)$ and $R_{sc}(Re_0, slab)$ in other channels highlight the radiance uncertainty due to the slab-cloud assumption. The RMSE in $R_{sc}(Re_0, slab)$ relative to $R_{mix}(Re, IWC)$ are shown by the red dashed curve in Fig. 3 (a).

Fig. 3 (a) reveals that the slab-cloud assumption cannot fully account for spectral variations of cloud emission. The assumption leads to a spectrally tilted mean radiance bias as shown by the red curve in Fig. 3 (a). We note that this tilted pattern is related to the spectrally dependent extinction coefficients, which is affected by effective radius (variation in $Re_{opt}$), so that radiances at different wavenumbers are contributed by cloud emission at varying heights, which is in turn affected by the vertical distribution of ice mass. Therefore, the clear-cut cloud boundary in the slab-cloud and a constant effective radius ($Re_{opt}$) collectively contribute to the radiance bias shown by a dashed blue curve in Fig. 3 (a). The RMSE of $R_{sc}(Re_0, slab)$ shows a minimum of around 0.2 K in the mid-infrared window and a maximum over 4 K at the high wavenumbers (over 2000 cm$^{-1}$). This RMSE spectrum is also calculated adopting an AIRS-like spectral response function to represent the radiance uncertainty induced by the slab-cloud assumption in the retrieval described in Section 2.3 and is denoted as $\varepsilon_{slab}$.

## 2.3 Retrieval Algorithm

The cloud-assisted retrieval proposed by Feng and Huang (2018) is an optimal estimation method (Rodgers, 2000) that retrieves atmospheric states above clouds using infrared spectral radiance. Similar to Eq.1 in Feng and Huang (2018), we express the

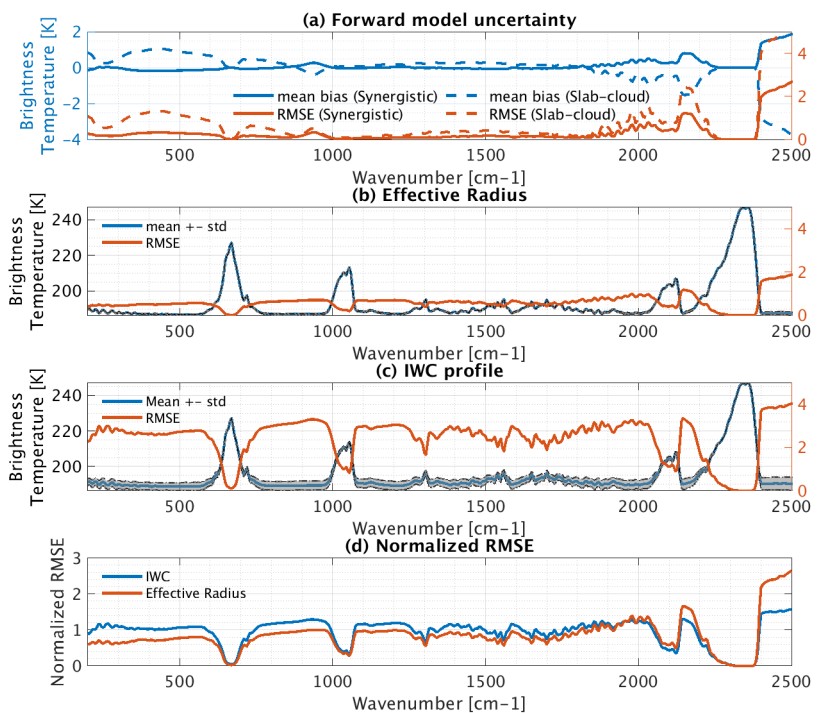

**Figure 3.** Effects of variations in tropical deep convective clouds on infrared radiance spectra from 200 to 2500 $\mathrm{cm}^{-1}$ at a 5 $\mathrm{cm}^{-1}$ resolution. (a) The mean bias (blue, left y-axis) and RMSE (red, right y-axis) in $R_{sc}(Re_{opt}, IWC)$ (solid) and $R_{sc}(Re_{opt}, slab)$ (dashed), respectively, relative to $R_{mix}(Re, IWC)$. (b) The mean radiance spectrum of $R_{sc}(Re_{opt}, IWC_0)$ (blue, left y-axis) and its STD (grey area). Red curves (right y-axis) are the RMSE in $R_{sc}(Re_0, IWC)$ relative to $R_{sc}(Re_{opt}, IWC)$. (c) The mean radiance spectrum of $R_{sc}(Re_0, IWC)$ (blue, left y-axis) and its STD (grey area). Red curves (right y-axis) are the RMSE in $R_{sc}(Re_{opt}, IWC_0)$ relative to $R_{sc}(Re_{opt}, IWC)$. (d) The RMSE in $R_{sc}(Re_0, IWC)$ (blue) and $R_{sc}(Re_{opt}, IWC_0)$ (red), respectively, relative to $R_{sc}(Re_{opt}, IWC)$ normalized by the spectral mean.

relation between the observation vector, $y$, and the state vector, $x$, as follows:

$$y = F(x_0) + \frac{\partial F}{\partial x}(x - x_0) + \varepsilon \tag{1}$$

$$= y_0 + K(x - x_0) + \varepsilon \tag{2}$$

Following a similar definition to Feng and Huang (2018), the state vector includes temperature $x_t$ and the logarithm of specific humidity, $x_q$, in 67 model layers. $x_0$ refers to the first guess of the state vector, which is the mean of the *a priori*. $y$ contains the infrared radiance observations, $y_{rad}$. $F$ is the forward model that relates $x$ to $y$. Here, the forward model is the radiative transfer model, MODTRAN 6.0, configured with the spectral response function of the AIRS instrument. The forward model can be linearly approximated by the Jacobian matrix $K$, which is iteratively computed at every time step. $\varepsilon$ is the measurement error that includes radiometric uncertainties of the instrument and forward model error. The forward model error comes from the radiative transfer algorithm used by the forward model and inputs to the forward model. Because the line-by-line algorithm of MODTRAN has been validated against LBLRTM (Berk and Hawes, 2017), we consider the forward model error to mainly arise from the uncertainties in the inputs, namely the cloud assumptions in the radiative transfer simulation which is evaluated in Section 2.2.1. Other uncertainties in the forward model calculations are neglected.

Following the optimal estimation method (Rodgers, 2000, Eq.5.16), an estimate of $x$, $\hat{x}$, is expressed as:

$$\hat{x} = x_0 + GK(x - x_0) + G(y - Kx) \tag{3}$$

$$G = S_a K^T (K S_a K^T + S_\varepsilon)^{-1} \tag{4}$$

where $S_a$ and $S_\varepsilon$ are the covariance matrix of the state vector as given by the *a priori* dataset and that of the error in the observation vector, respectively. $S_\varepsilon$ is set to be a diagonal matrix because the observation errors in different channels are considered to be uncorrelated.

$\hat{x}$ can then be iteratively solved through:

$$\hat{x}_{i+1} = x_0 + (K_i^T S_\varepsilon^{-1} K_i + S_a^{-1})^{-1} K_i^T S_\varepsilon^{-1} [y - F(\hat{x}_i) + K_i(\hat{x}_i - x_0)] \tag{5}$$

where the subscript $i$ refers to the $i$th iteration step.

The equations described above are adopted from Feng and Huang (2018), where the state vector $x$ includes temperature and the logarithm of specific humidity. For comparison, we adopt the slab-cloud retrieval scheme of Feng and Huang (2018) as described above and refer to the result as the slab-cloud retrieval in the following. The only difference from Feng and Huang (2018) is in the $S_\varepsilon$. While $S_\varepsilon$ in Feng and Huang (2018) is the square of radiometric noise of AIRS instrument, $S_\varepsilon$ in this study for slab-cloud retrieval contains the sum of the square of radiometric noise and the square of $\varepsilon_{slab}$, as schematically depicted by the red dashed curve in Fig. 3(a), to account for radiance uncertainties induced by slab-cloud assumption. Because $\varepsilon_{slab}$is relatively small, especially at absorption channels, we find that adding off-diagonal correlations to $S_\varepsilon$ does not improve the retrieval quality significantly. Therefore, $S_\varepsilon$ keeps its diagonal form.

We further examine whether the addition of $\varepsilon_{slab}$ masks spectral signals from atmospheric variations. Figure 1 depicts that strong cooling and hydration appears above overshooting DCCs near the cyclone center (141°E, 16°N). We denote

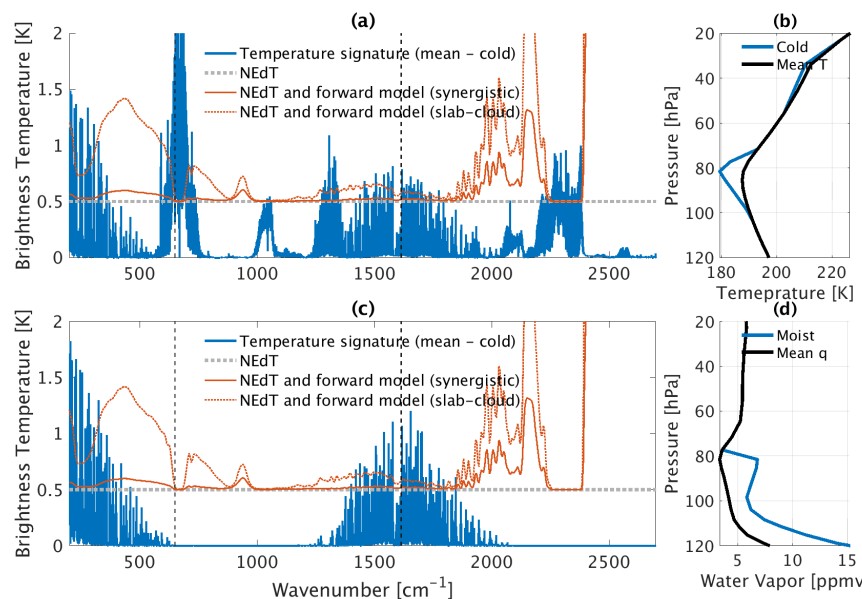

**Figure 4.** Spectral signals of above-storm atmospheric variations in (a) temperature and (c) water vapor from 200 to 2500 cm$^{-1}$. The signals are obtained by differencing the upwelling radiance spectra at the TOA simulated from the mean of all profiles (black curves in panels b and d) and radiance spectra simulated from the mean of the profiles with overshooting convective clouds near the cyclone center (blue curves in panel b and d). These signals are shown at a spectral resolution of 0.1 cm$^{-1}$. In a and c, the light grey dotted lines denote the NEdT of 0.5 K (characterizing the AIRS instrument at cold scene temperature). The red solid lines denote the uncertainties combining the NEdT and $\varepsilon_{synergistic}$ and the red dotted lines denotes those combining the NEdT and $\varepsilon_{slab}$, which are convoluted at 5 cm$^{-1}$ spectral intervals in this plot. The AIRS spectral range used in this study is between 649.6 cm$^{-1}$ and 1613.9 cm$^{-1}$ and is marked by dark grey dashed lines.

the mean profiles of this region as $t_{cold}$ and $q_{mois}$ for temperature and water vapor, respectively, which are shown in Fig.4 (b,d) blue curves. A set of radiative transfer calculations is conducted to obtain $R_{sc}(Re_0, t_0, q_0)$, $R_{sc}(Re_0, t_{cold}, q_{mois})$ and $R_{sc}(Re_0, t_{cold}, q_0)$ at 0.1 cm$^{-1}$ spectral resolution, using an effective radius of 34 $\mu$m and a randomly selected IWC profile (cloud top at 100 hPa) of this region. The spectral signals of temperature and water vapor are then obtained from $R_{sc}(Re_0 t_0, q_0) - R_{sc}(Re_0, t_{cold}, q_0)$ and $R_{sc}(Re_0, t_0, q_0) - R_{sc}(Re_0, t_0, q_{mois})$, respectively. The strength of signals under

different spectral specifications was examined by Feng and Huang (2018) and is not repeated here. The spectral signals are then compared to radiance uncertainties in Fig. 4. The spectral range used in the retrieval tests (between 649.6 and 1613.9 cm$^{-1}$) are enveloped by grey dotted lines in Fig. 4.

     Figure 4 shows that the radiance uncertainty from the slab-cloud assumption, $\varepsilon_{slab}$, does not completely obscure the signal of temperature or water vapor. In the $CO_2$ and water vapor channels, where the signal is the strongest, the TOA radiance

spectra are not as sensitive to cloud emission due to strong atmospheric attenuation at these channels. $\varepsilon_{slab}$ becomes greater in the wings of absorption channels, where the signals are already masked by the instrument NEdT of $\sim$ 0.5 K.

### 2.3.1 Synergistic method

The radiance uncertainty due to the slab-cloud assumption, $\varepsilon_{slab}$, can be largely eliminated by incorporating collocated observations of cloud profiles, from active sensors (CloudSat-CALIPSO) along the same track as the hyperspectral infrared sounder (such as AIRS). Instead of simply prescribing the cloud profile from the active sensors in the forward model, motivated by the work of Turner and Blumberg (2018), we include relevant cloud variables in a synergistic retrieval. Turner and Blumberg (2018) demonstrated that additional observation vectors, such as atmospheric and cloud profiles from other instruments or NWP products, can improve convergence in cloudy scenes and the precision of the retrieval by constraining the posterior uncertainty. Following this idea, the observation vector $y$ in Eq.1 is formulated as: $[y_{rad}, y_{other}]$, where $y_{rad}$ contains the infrared radiance observations, and $y_{other}$ includes elements other than radiance observations; we refer to the latter as the additional observation vector. Collocated cloud observations are added to $y_{other}$ to mimic cloud properties obtained from the DARDAR-Cloud product. At every iteration step (Eq. 5), IWC and effective radius are included in the state vector $x$ and are updated along with the temperature and humidity profiles.

In this simulation experiment, the observation vector for IWC, $y_{iwc}$, is set to be the natural logarithm of the IWC profile to account for IWC variations from $O(10^{-5})$ to $O(1)$ g/m$^3$ and to avoid negative values. Uncertainties in IWC measurements are estimated by averaging the posterior uncertainty of IWC, provided by the DARDAR-Cloud product for every footprint, in the OT-DCC profiles identified in Section 2.2.1. This estimated precision is denoted as $\varepsilon_{iwc}$, which corresponds to a roughly $20\%$ uncertainty in IWC at vertical levels near the tropopause. Then, we account for the IWC observation uncertainty by randomly perturbing the $y_{iwc}$ so that the $y_{iwc}$ deviates from the true state by an error that has an STD of $\varepsilon_{iwc}$. As mentioned in Section 2.2, the effective radius profiles of the test set are sampled from the DARDAR-Cloud product. However, we do not intend to retrieve the effective radius profile because mid-infrared radiance spectra are not sensitive to layer-to-layer variations in effective radius, as found in Section 2.2.1. Instead, the effective radius for representing the spectral emission of an entire cloud column is retrieved, which is the same as $Re_{opt}$ defined in Section 2.2.1. The true $Re_{opt}$ is obtained by approaching the true radiance spectra through iteration. The observation vector for $Re_{opt}$, $y_{Re_{opt}}$, is constructed by randomly perturbing the effective radius value at the layer where the optical depth measured from the cloud top reaches unity ($Re_{\tau=1}$) with an uncertainty of $5$ $\mu$m. We note that this prescribed uncertainty is larger than the typical value in the DARDAR-Cloud product ($1.6$ $\mu$m) to account for sampling differences between the instruments. Because the satellite-measured infrared radiance spectra are most sensitive to cloud emission near the cloud top, only the top $1.5$ km of the IWC profile in $y_{iwc}$ is kept, which corresponds to six model layers in the radiative transfer calculations.

The state vector, $x_{iwc}$, contains six layers of the logarithm of IWC at the same model layers as $y_{iwc}$. Note that $x_{iwc}$ and $y_{iwc}$ are not required to have the same vertical resolution; in practice, the vertical resolution of $y_{iwc}$ can be much finer than that of the model layers. The first guess and covariance matrix of $x_{iwc}$ are calculated using the same *a priori* dataset described in the previous section, although cross-correlations between IWC and other atmospheric variables are neglected. Consequently, the forward model for relating $x_{iwc}$ to $y_{iwc}$ is a matrix that linearly interpolates the pressure level of $x_{iwc}$ to match the level of $y_{iwc}$ (Eq. 6 in Bowman et al., 2006). The *a priori* in $x_{Re_{opt}}$ is 34 $\mu$m with an uncertainty range of 11 $\mu$m. The diagonal

elements of $S_\varepsilon$ for $y_{iwc}$ and $y_{Re_{opt}}$ are set by conservatively quadrupling the square of the uncertainty ranges of the variables (20% for IWC and 5 $\mu m$ for $Re_{opt}$).

In this synergistic retrieval framework, cloud optical property inputs to the forward model are considered to be the major source of uncertainties of the forward model. While $IWC$ and $Re_{opt}$ are retrieved states, uncertainties in other cloud variables should be included in the forward model error that is quantified by $\varepsilon_{synergistic}$ in Section 2.2.1. Therefore, the $S_\varepsilon$ for $y_{rad}$ in the synergistic method contains the sum of the square of radiometric noise and the square of $\varepsilon_{synergistic}$. As shown in Fig. 3 and 4, $\varepsilon_{synergistic}$ is much smaller compared to $\varepsilon_{slab}$ and spectral signals from temperature and water vapor. It is also smaller than the spectral RMSE caused by $IWC$ and $Re_{opt}$ with a distinct shape in mean biases (see Fig. 3).

### 2.3.2 Additional atmospheric observations

Besides the cloud observations, other products that provide collocated atmospheric profiles can be useful in improving the precision of the posterior estimation. These additional products may include atmospheric observations from other instruments that are in the same satellite constellation as the hyperspectral infrared sounder. It can also come from reanalysis products, which typically do not assimilate cloudy infrared radiances in operation. In this study, we investigate the effect of incorporating coincident reanalysis products by adding an observation vector $y_{atm}$, which contains the temperature and the logarithm of specific humidity at a later time step: 810 minutes after the initial time, in the GEM simulation. This arbitrary choice of the simulation time step is to represent the potential quantitative differences in temperature, humidity, and cloud fields between a reanalysis product and the true state.

Distributions of retrieval variable fields are shown in Fig. 6. As inferred by the brightness temperature, the massive spatial coverage of DCCs is evident at the time step used as the 'Truth' (410 minutes after the initial time in the GEM simulation). At the later time step (810 minutes), the atmospheric data used as $y_{atm}$ are taken from the same locations but deviate from the 'Truth' as they are not directly above convective overshoots at this later time step. The RMSE between atmospheric profiles from the two time steps (410 and 810 minutes) defines the uncertainties in $y_{atm}$. To be conservative, the uncertainty of $y_{atm}$ is set by quadrupling the square of the RMSE in the corresponding diagonal elements of $S_\varepsilon$.

## 3 Results

Four retrieval cases are designed to assess the retrieval performance following different strategies. Among them, Cases 1 and 2 use the slab-cloud method; and Cases 3 and 4 use the synergistic method that incorporates cloud observations. Cases 2 and 4 differ from Cases 1 and 3 in that they add $y_{atm}$ in the retrieval. The components of the state and observation vectors for the four cases are listed in Table 1. An additional case 5 is performed, which follows the same optimal estimation framework without using infrared radiances $y_{rad}$ as in Case 4. It is expected to converge to *a posterior* state that is jointly determined by the *a priori* profile, $y_{iwc}$, and $y_{atm}$. Therefore, the statistical differences between Case 4 and 5 ascertain the improvements attributable to infrared radiances (as opposed to other sources of information). Case 5 is relatively uniform in space and it is therefore not included in the figures but listed in Tables 1 and 2 for comparison. Following the framework of this simulation

experiment, retrievals are then performed for the 100-profile test set, using synthetic radiance observations ($y_{rad}$) generated in Section 2.2, IWC ($y_{iwc}$) and effective radius ($y_{Re_{opt}}$) described in Section 2.3.1, and additional atmospheric product ($y_{atm}$) constructed in Section 2.3.2. Retrieval performances are examined by mean biases and RMSE in Fig. 5 and 6. Retrieved temperature, water vapor, and IWC profiles are also compared to the first guess, observation constraints ($y_{other}$), and the truth in Fig. 7 for two samples from the test set.

We next examine the DFS [degrees of freedom for signal, (Rodgers, 2000)] of temperature, water vapor, IWC, and effective radius in the four retrieval cases (Table 3.1). DFS is defined as the trace of the averaging kernel A, which relates the retrieved state $\hat{x}$ to the true state $x_0$ , as derived from Equation 3.5 at the end of the iteration:

$$\hat{x} - x_0 = A(x - x_0) \tag{6}$$
$$A = (K^T S_\varepsilon^{-1} K + S_a^{-1})^{-1} K^T S_\varepsilon^{-1} K \tag{7}$$

While all observation vectors are used in the retrieval, only the radiance observation, $y_{rad}$, is included to calculate the DFS, so that a higher DFS indicates higher information content brought by $y_{rad}$ alone. Because the DFS depends on the cloud distribution, the DFS shown in Table 1 is averaged over the 100-profile test set.

Although $\varepsilon_{slab}$ does not mask the observable signals in Fig. 4 (a,b), the DFS for temperature increases from 3.15 (Case 1) to 3.6 (Case 3) when the synergistic method is adopted. The improved DFS highlights the strong sensitivity of the synergistic method to temperature near the cloud top. In comparison, the slab-cloud method fails to fully capitalize on information near the cloud top, as it neglects contributions from the vertical layers around the assumed sharp cloud boundary. Therefore, the synergistic method is expected to achieve a better result for temperature.

Moreover, significant DFS values are found for IWC (1.94 out of 6, on average) and effective radius (0.66 out of 1, on average). The DFS confirms the sensitivity of infrared radiances to the IWC profile and effective radius near the cloud top, which is consistent with large radiative perturbation caused by varying IWC (Fig. 3 (c-d)) based on the DARDAR-Cloud product. The DFS for IWC varies from 0.96 to 2.71 in the test set, depending on the optical depth near the cloud top. Low ice density near the cloud top leads to a higher DFS for IWC and effective radius. For example, the DFS for IWC increases from 1.30 in Fig. 7 (c) to 2.63 in Fig. 7 (f), because thermal emission from lower levels can be transmitted through the topmost cloud layer. In the meantime, the DFS for effective radius increases from 0.04 to 0.66, because the thermal emission is more sensitive to the spectral shape of extinction coefficients induced by effective radius (as depicted in Fig. 3 (c)) when optical depth is small. Overall, the DFS values suggest that a synergistic method can improve the precision of IWC and effective radius measurements relative to collocated cloud products alone.

Retrieval performance is evaluated through the mean bias and RMSE in temperature, humidity, and IWC between the retrieved profiles and the truth, as shown in Fig. 5. The retrieval performance is also evaluated with regard to these quantities at selected levels and with regard to CIWV integrated from 110 to 70 hPa.

**Table 1.** State vector and observation vector of four cases of retrieval strategies. Case 5 is a posterior estimation of the state vector from a combination of $y_{atm}$, $y_{iwc}$, $y_{Re_{opt}}$, and *a priori*. DFS is compared to the number of vertical layers of the state vector. The DFS is counted from 130 hPa to 13.5 hPa for temperature and water vapor (20 model layers).

| | $x$ | $y$ | DFS |
|---|---|---|---|
| | | **Slab-cloud** | |
| Case 1 | $x_t, x_q$ | $y_{rad}$ | $t$: 3.15, $q$: 0.69 |
| Case 2 | $x_t, x_q$ | $y_{rad}, y_{atm}$ | Same as Case 1 |
| | | **Synergistic** | |
| Case 3 | $x_t, x_q, x_{iwc}, x_{Re_{opt}}$ | $y_{rad}, y_{iwc}, y_{Re_{opt}}$ | $t$: 3.6, $q$ :0.74, $IWC$ :1.94, $Re_{opt}$: 0.65 |
| Case 4 | $x_t, x_q, x_{iwc}, x_{Re_{opt}}$ | $y_{rad}, y_{atm}, y_{iwc}, y_{Re_{opt}}$ | same as Case 3 |
| Case 5 | $x_t, x_q, x_{iwc}, x_{Re_{opt}}$ | $y_{atm}, y_{iwc}, y_{Re_{opt}}$ | \ |

**Table 2.** Performance assessments of four cases of retrieval strategies, in comparison with the prior, the observation vector, and Case 5.

| | $t$ [K] at 81 hPa | | $q$ [ppmv] at 81 hPa | | CIWV [g/m$^2$] from 110 to 70 hPa | | IWC [g/m$^3$] at 90 hPa | | Re [$\mu$m] | |
|---|---|---|---|---|---|---|---|---|---|---|
| | Bias | RMSE | Bias | RMSE | Bias | RMSE | Bias | RMSE | Bias | RMSE |
| prior | 6.8 | 7.1 | -1.8 | 1.5 | -0.30 | 0.34 | 0.0014 | 0.0413 | -5 | 11 |
| $[y_{atm}, y_{iwc}]$ | 8.1 | 10.6 | -1.7 | 2.3 | -0.17 | 0.24 | 0.0029 | 0.0096 | -0.8 | 4.8 |
| | | | | | **Slab-cloud** | | | | | |
| Case 1 | 0.2 | 4.5 | -1.8 | 2.4 | -0.29 | 0.36 | \ | \ | \ | \ |
| Case 2 | 1.1 | 4.1 | -0.7 | 1.0 | -0.11 | 0.16 | \ | \ | \ | \ |
| | | | | | **Synergistic** | | | | | |
| Case 3 | 0.0 | 2.4 | -1.8 | 2.3 | -0.2 | 0.30 | -0.0029 | 0.0075 | 2.4 | 4.3 |
| Case 4 | 0.8 | 2.7 | -0.8 | 1.1 | -0.09 | 0.16 | 0.0015 | 0.0056 | 1.8 | 3.7 |
| Case 5 | 2.7 | 4.9 | -1.8 | 2.4 | -0.18 | 0.24 | 0.0029 | 0.0096 | -0.8 | 4.8 |

## 3.1 Slab-cloud retrieval

Improving upon Feng and Huang (2018), Case 1 accounts for the radiance uncertainties due to the slab-cloud assumption, while Case 2 further incorporates additional atmospheric constraints to improve the precision of the method.

The results of Case 1 are shown as red solid curves in Figures 5 and 7. The major improvement in Case 1, compared to the prior (blue solid curves), is the temperature profile from 100 to 75 hPa. Although DFS for water vapor reaches 0.69, Case 1 does not provide much improvement from the first guess in water vapor.

Case 2 improves from Case 1 owing to the information carried by the additional atmospheric constraints, $y_{atm}$. Case 2 is represented by the red dotted curves in Figures 5 and 7. It approaches the true state better than Case 1, despite warm and dry biases in the first guess and $y_{atm}$ (See Fig. 5 (a,c)). Notably, it increases the retrieved water vapor concentration by around 1 ppmv on average and reduces the RMSE from 2.4 ppmv to 1.0 ppmv, as shown in Fig. 5 (c,d) and Table 3.2. For the CIWV, Case 2 reduces the RMSE by half when compared to Case 1.

To demonstrate how well the retrieved atmospheric field represents the spatial variability in the true state (Fig. 1), namely a moister and colder UTLS region in the cyclone center compared to the south of the domain, the distributions of water vapor, temperature, and CIWV are presented in Fig. 6. It shows that the 'true' spatial patterns are well reproduced by the Case 2 retrieval.

Furthermore, individual profiles from two clusters of overshooting DCCs, which include the DCCs near the cyclone center and those in the south of the domain, are randomly selected to investigate how well the retrieval reproduces the spatial variability in temperature and water vapor. The all-sky optical depths from TOA and IWC profiles for the two locations are shown in Fig. 7 (c,d). The retrievable signals mainly come from the atmospheric column above thick cloud layers, i.e., where optical depth is less than 2 (only 13.5% of the infrared emission is transmitted through this cloud layer).

Figure 7 (a-c) shows results for a location close to the cyclone center. At this location, the slab-cloud method prescribes the cloud layer to be located at the cold-point due to the strong cloud emission. Atmospheric anomalies above 86 hPa have an impact on TOA infrared radiances. Around 80 hPa, the truth profile that we aim to retrieve is around 8 K colder than the prior and nearly 3 ppmv moister. While the result from Case 1 overcomes the bias in temperature, it increases the water vapor over a broad vertical range which, as explained by Feng and Huang (2018), is due to the strong smoothing (smearing) effect of the averaging kernel in this case. In comparison, Case 2 correctly produces a peak moistening around 80 hPa, while keeping a retrieved temperature profile similar to Case 1.

Figure 7 (d-f) shows the results in a location in the southern part of the domain, where the slab-cloud method prescribes the cloud layer at 95 hPa. At this location, the cloud emission from the top 1.5 km cloud layer affects infrared radiances strongly, which can be inferred from the optical depth (Fig. 7 (f)), leading to a large radiance residual that cannot be addressed under the slab-cloud assumption. Therefore, Case 1 fails to improve upon the prior. Case 2 leads to a moister posterior compared to the prior owing to the addition of $y_{atm}$. However, Case 2 fails to update the temperature profile above the cloud layer. Instead, it approaches $y_{atm}$ in lower altitudes, leading to an unrealistic vertical oscillation in temperature near 100 hPa.

## 3.2 Synergistic method

Using the synergistic method, Case 3 becomes more sensitive to water vapor and temperature compared to Case 1, as indicated by the reduced RMSE in Table 3.2 and a closer match between the retrieved field and the true state in Fig. 6. It retrieves higher water vapor concentrations from 110 to 70 hPa in comparison with Case 1. Owing to the radiative emission from in-cloud

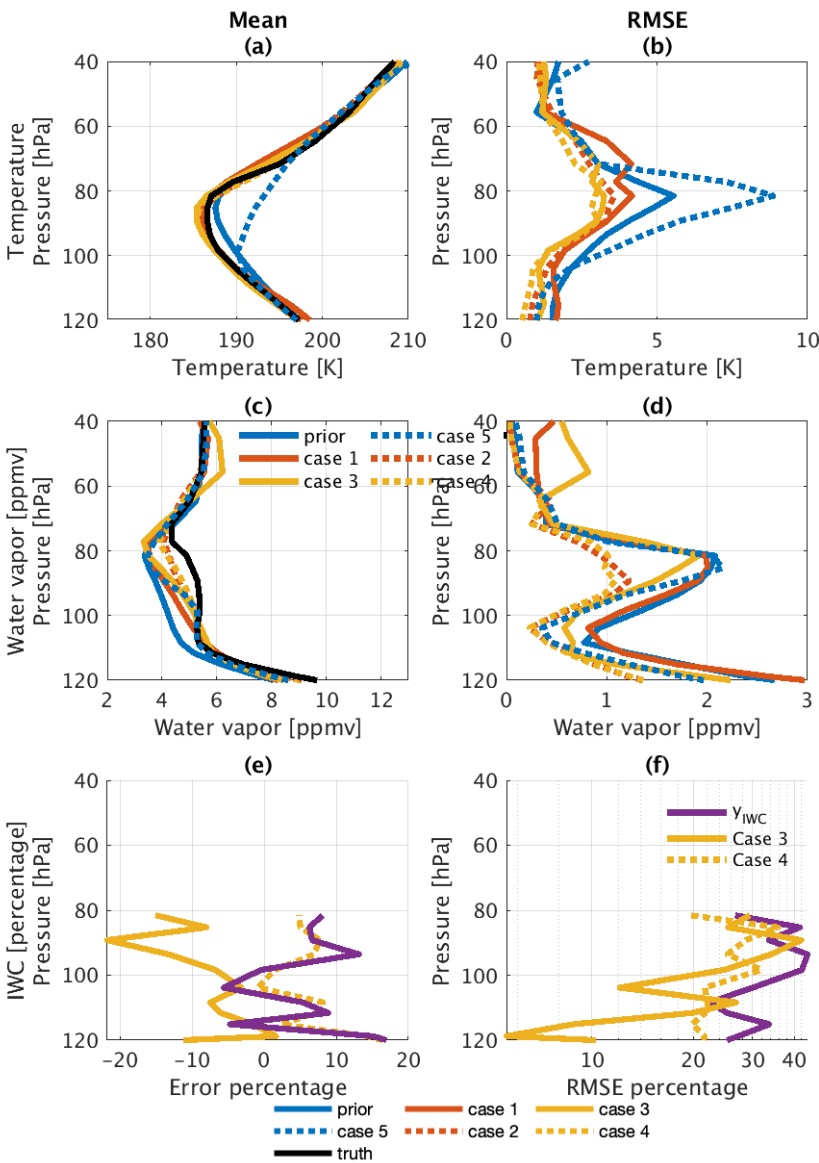

**Figure 5.** The mean and RMSE of temperature (a,b), water vapor (c,d) profiles from the four cases of retrieval strategies. Percent mean bias (e) and RMSE (f) of IWC profiles. Blue curves show the bias and RMSE in the prior. Retrieval cases using the slab-cloud method are marked by the red curves (case 1 in solid and case 2 in dotted curves), while the retrievals using the synergistic method are marked by yellow curves (case 3 in solid and case 4 in dotted curves).

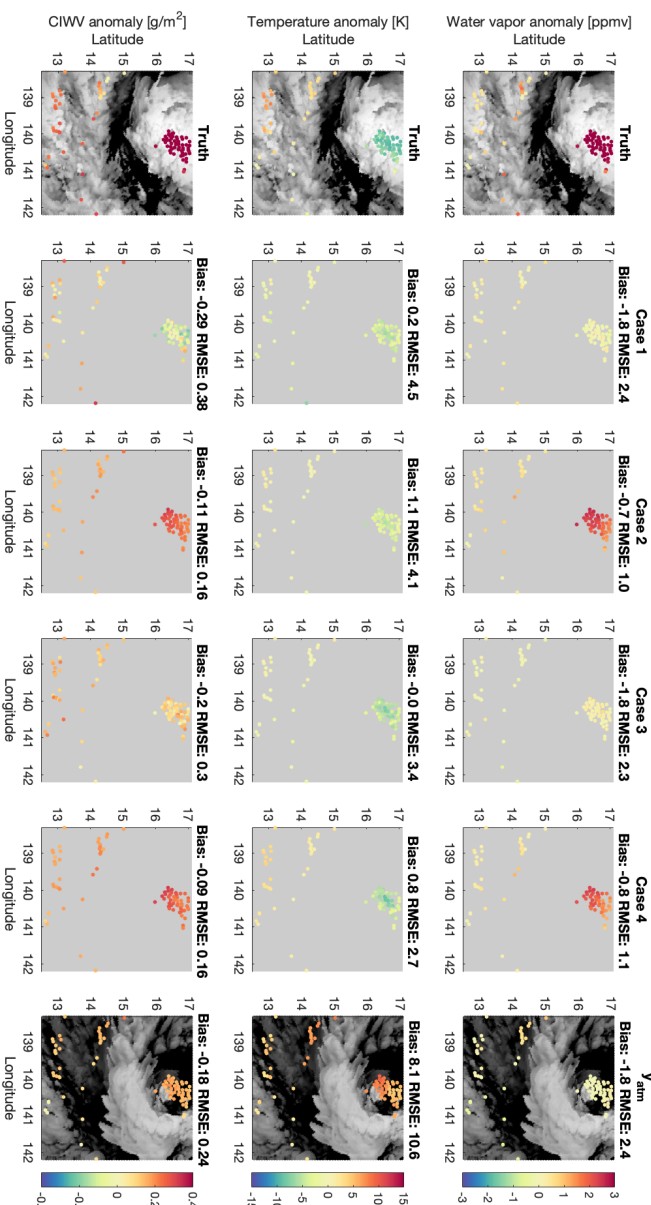

**Figure 6.** Horizontal distributions of the anomalies, defined as the deviation from $x_0$, in water vapor (in the units of $\mathrm{ppmv}$, upper panels), temperature (in the units of $\mathrm{K}$, middle panels) at 81 hPa, and column integrated water vapor between 110 and 70 hPa (in the units of $\mathrm{g/m^2}$, lower panels). The true states are shown in the first row, with background grey shaded for $BT_{1231}$. The second to fifth rows show retrieved results from the four cases of retrieval strategies described in Table 3.1. The sixth-row shows distribution of the additional observation vector, $y_{atm}$, incorporated in the retrievals of Cases 2 and 4. This additional atmospheric constraint, $y_{atm}$, is taken from the model fields 810 minutes after the initial simulation time step.

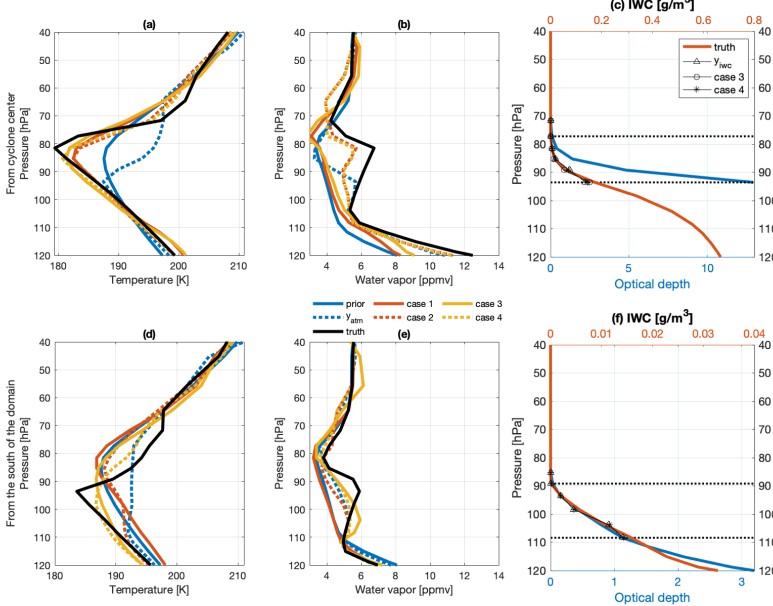

**Figure 7.** (a,d) Temperature and (b,e) water vapor profiles of first guess (blue solid), truth (black solid), and posterior results (red and yellow curves represent the slab-cloud and synergistic methods, respectively, while the solid curves are cases without $y_{atm}$ and dotted curves include $y_{atm}$) for two profiles from the test set. (c,f) True IWC (red curve, corresponding to the upper x-axis) and all-sky optical depth from the TOA (blue curve, corresponding to the lower x-axis) of the two selected profiles. $y_{iwc}$ and retrieved IWC profile are marked by triangles, circles (case 3), and asterisks (case 4), respectively. Dotted black lines mark the vertical ranges of ice cloud information included in $y_{iwc}$ and $x_{iwc}$, while ice clouds in the lower vertical levels are prescribed to be the same as the cloud observation.

layers between 110 and 95 hPa, Cases 3 and 4 become sensitive to the temperature profile near the cloud top. Hence, Cases 3 and 4 reduce the RMSE compared to other cases.

The advantage of the synergistic method, especially when IWC near the cloud top is relatively small, is illustrated in Fig. 7 (d-f). At this location, the radiative signal from the moistening near the cloud top can be transmitted to the TOA. As a result, Case 3 approaches the true cloud-top temperature much better than Cases 1 and 2 (Fig. 7 (a,d)). It also produces higher water vapor compared to Case 1 (Fig. 7 (e)). Case 4 further benefits from $y_{atm}$ which constrains the profile in the vertical ranges below 110 hPa and above 80 hPa. Case 4 overcomes the warm bias around 90 hPa in $y_{atm}$ and the first guess. It also reproduces the oscillating temperature feature in Fig. 7 (d).

Owing to the sensitivity to IWC and $Re_{opt}$ as suggested by DFS and Fig .3, the synergistic method can improve upon the collocated cloud observations by reducing RMSE in IWC profile and in $Re_{opt}$ (Fig. 5 and Table 2). Although adding the $y_{atm}$ does not significantly improve the retrieval performance in case 4, it stabilizes the iterative retrieval process by constraining uncertainties in temperature. The improvement from infrared spectra and the addition of $y_{atm}$ is desirable to reduce measurement uncertainties due to sampling differences between active sensors and the infrared instrument.

While the improvement in Cases 2 and 4 shows the advantage of including additional atmospheric products, $y_{atm}$, one caveat is in the proper evaluation of the uncertainty range, which is included in the covariance matrix of the observation vector. This is important as the uncertainty range in $y_{atm}$ constrains the posterior uncertainty range of the retrieval at each vertical level. In this study, we account for the difficulties in evaluating $S_\varepsilon$ by increasing the RMSE in $y_{atm}$, so that the square root of $S_\varepsilon$ of $y_{atm}$ is equivalent to a doubling of RMSE shown by the blue dotted line in Fig. 5 (b,d).

Although the additional measurement vector, $y_{atm}$, itself does not contain the spatial variability pattern as seen in Fig. 6, the corresponding covariance in $S_\varepsilon$ properly accounts for its variability (uncertainty) by prescribing a large value around 80 hPa but smaller values at other vertical locations. Therefore, it increases confidence in the posterior at levels where the thermodynamic variables are relatively constant. The increased confidence in turn enhances the degrees of freedom in the ranges around 80 hPa, where the warm and dry signals mainly come from. Therefore, even though $y_{atm}$ itself deviates from the true state, including $y_{atm}$ in optimal estimation can still improve the posterior estimation. In practice, uncertainty in atmospheric products can be estimated by inflating the precision of the product to account for sampling size differences through comparison with NWP models and collocated observations.

## 4   Conclusion and Discussion

Sounding UTLS thermodynamic conditions has long been a challenge. A simulation experiment has been conducted to simulate hypothetical radiance observations of AIRS by integrating a NWP model and a radiative transfer model, MODTRAN 6.0. By conducting the simulation experiment, this study evaluates the capability of existing hyperspectral infrared sounders in detecting temperature and humidity fields above convective storms. Our focus is to investigate and constrain the uncertainties induced by clouds. Two retrieval methods are tested, including a slab-cloud method that uses mainly the infrared radiance measurements (i.e., AIRS) and a synergistic method that combines cloud products from collocated active sensors (i.e., DARDAR-Cloud).

First, we find that a radiative transfer model can simulate the TOA mid-infrared radiance spectra above tropical deep convective clouds fairly accurate (RMSE around 0.1 K, characterized by $\varepsilon_{synergistic}$) by assuming constant cloud optical properties (per unit mass) in all vertical layers of a cloud column. Uncertainties in the infrared radiance spectra mainly comes from variations in IWC profile and column-to-column variations in effective radius (Fig. 3). The uncertainties are largest in window channels and weak absorption channels because they are sensitive to cloud emission. The slab-cloud assumption locates a clear-cut cloud top that matches the brightness temperature of the window channel. This assumption alleviates, but does not fully eliminate, the cloud effect on the radiance spectrum (Fig. 3 (a)). The remaining radiance uncertainty is accounted for in the retrieval framework of this study and is found to not significantly obscure the temperature and humidity signals in the retrieval. Therefore, the cloud-assisted retrieval as proposed by Feng and Huang (2018) is affirmed to improve the sounding of UTLS temperature and water vapor compared to prior knowledge. However, this retrieval neglects information content from the in-cloud atmosphere. As a result, it may lead to biases in individual temperature profiles. For example, as shown in Fig. 7 (c), the slab-cloud retrieval fails to reproduce oscillating temperature anomalies, although it still detects anomalous moistening

above convective storms. Although not explicitly discussed here, a similar OE framework adopting the slab-cloud assumption is expected to detect moistening anomalies when applied to other hyperspectral infrared sounders, e.g., IASI and CrIS, due to their similar spectral specifications to AIRS.

Second, we find that the synergistic method, especially after incorporating additional atmospheric constraint, $y_{atm}$, is sensitive to temperature, water vapor, the IWC profile, and column-to-column variation in effective radius. It substantially reduces the RMSE in temperature from 7.1 to 2.7 K compared to the prior. It also reduces the RMSE in column integrated water vapor by half. This method can capture strong moistening features in individual profiles (as shown by Fig. 7 (b)) and detect oscillating temperature anomalies (as shown by Fig. 7 (c)). The retrieved temperature and humidity fields by synergistic approach best

match the true horizontal distribution patterns on a fixed pressure level (Fig. 6). Moreover, owing to the sensitivity of infrared radiance spectra to cloud properties, the synergistic method is able to improve IWC and effective radius ($Re_{opt}$) relative to collocated active cloud observations.

In conclusion, our study suggests that the synergistic method holds promise for using hyperspectral infrared radiance and cloud profiles from the existing instruments (AIRS, CloudSat, and CALIPSO) to retrieve UTLS temperature and water va-

por distributions above deep convective clouds. As discussed in Feng and Huang (2018), the sensitivity to water vapor and cloud microphysics properties (see Section 2.2.1) can be further improved by including the far-infrared coverage provided by future instruments, e.g., FORUM and TICFIRE. While a limited number of samples is available for applying the synergistic retrieval, instruments in geostationary orbit, such as IRS (Infrared Spectrometer) and GIIRS (Geostationary Interferometric Infrared Sounder) (Schmit et al., 2009; Holmlund et al., 2021), can greatly increase collocation with other space-borne ac-

tive sensors over convective regions. Such an approach may also benefit the understanding of convective impacts by providing time-continuous observations (Li et al., 2018) in future research. The ability of the synergistic method to leverage hyperspectral infrared observations to improve the NWP outputs ($y_{atm}$) also suggests the advantage of including cloudy-sky observations in global data assimilation systems, as performed by Okamoto et al. (2020).

*Data availability.*   Derived data supporting the findings of this study are available from JF on request. The data for assessing cloud-induced

uncertainties is openly available at http://dx.doi.org/10.17632/fy3gg7ch42.1.

*Author contributions.*   YH conceived the cloud-assisted retrieval idea; JF implemented this idea with improvements using the synergistic method. ZQ carried out the NWP simulation. JF and YH co-designed the simulation experiment and wrote this paper with contributions from ZQ.

*Competing interests.*   The authors declare that they have no conflict of interest.

*Acknowledgements.* We thank Sergio DeSouza-Machado, Quentin Libois, Jonathon Wright, Lei Liu, and an anonymous reviewer for their constructive comments. This work is supported by grants from the Canadian Space Agency (16SUASURDC and 21SUASATHC) and the Natural Sciences and Engineering Research Council of Canada (RGPIN-2019-04511). JF acknowledges the support of a Milton Leong Graduate Fellowship of McGill University. We thank Natalie Tourville for the public accessibility of the TC overpass dataset (https://adelaide.cira.colostate.edu/tc/). We thank ICARE Data and Services Center (http://www.icare-lille1.fr) and Dr. Julien Delanoë for access to the DARDAR product.

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
