# Peer review of "A simulation experiment-based assessment of retrievals of above-cloud temperature and water vapor using hyperspectral infrared sounder"

_Atmospheric Measurement Techniques, 2020_

## Referee Comment (RC1)

Review of « An observing system simulation experiment (OSSE)-based assessment of the retrieval of above-cloud temperature and water vapor using infrared hyper-spectrometers », by Jing Feng et al.

**General comments**

This paper is a follow-up of a previous study (Feng and Huang, 2018) that investigated the feasibility to retrieve stratospheric water vapor profiles above thick convective clouds using infrared hyperspectral observations. The present study explores in details the performances of various retrieval strategies (based on the optimal estimation method) and in particular the added value of using ancillary observations (ice cloud properties from the DARDAR synergistic product and atmospheric profiles in the vicinity of the scene) to retrieve water vapor, temperature, and ice water content profiles. Including vertical information on the distribution of ice near the top of the cloud greatly improves the retrievals compared to the slab-cloud hypothesis where an optically thick cloud is located at a single level in the retrieval algorithm, mostly when the top of the cloud exhibits low ice water contents. Using additional atmospheric information also improves the retrieval compared to using climatological a priori values.

The various retrieval strategies are generally well presented and detailed. The paper is well written and the figures appropriate. The manuscript, rather technical, perfectly fits for AMT journal. However the abstract lacks information and the introduction is probably too short to highlight the originality of the work compared to previous studies. A particular critical point is the use of the term "OSSE" to describe something that is probably more a standard retrieval, although tested on atmospheric fields originating from an NWP model. The multiplicity and relative complexity of the different cases described makes it sometimes difficult to follow the optimal estimation framework, but local rephrasing can certainly improve the readability. The results should probably put in a broader context to help the reader figure out whether the improvements for the various retrievals mentioned are significant, and how they compare to existing methods. This paper deserves being published in AMT, but the authors are encouraged to make their best to make the reading as easy as possible. Various suggestions are made below towards this.

**Specific comments**

1) The abstract is not an abstract so far. An abstract should indeed be a very short summary (very similar to the conclusion actually) of the work. It could be a bit longer, emphasizing for instance that generally sounders reject cloudy pixels (at least in NWP models). It should highlight quantitative results. An abstract should be self-sufficient and provides details : what instruments are used, how much are the performances improved etc. ? The abstract needs rewriting.

2) The term "OSSE" is used, in particular in the title, to define the type of experiment performed in this study. Although NWP model outputs are used to test the retrieval algorithms, the present study is not an OSSE, in the sense that no forecasts are performed with the NWP models. In general, an OSSE implies to compare two NWP forecasts, one of those including the assimilation of observations from a supplementary instrument compared to the other. The gain in forecast skill between both forecasts can then be attributed to the additional use of this new instrument. See for instance Arnold and Dey (1986).

3) Some descriptions of the retrieval algorithms are not very clear, in particular when it comes to the details of the optimal estimation method, with the definition of observation vectors, state vectors and covariance errors. In particular Case 5 experiment is not sufficiently clear. It is essential that the combination of the text and equations allows the reader to reproduce the algorithm. The same holds

for the definition of DFS relative to each state variable for instance. Likewise, the way atmospheric profiles are sometimes modified for sensitivity studies is not always clear (see examples below).

4) The study deliberately focuses on high thick clouds to probe UTLS atmospheric properties. However the title seems to be more general and does not exclude other types of clouds, including for instance low clouds or thin ice clouds. To make the paper more general it'd be useful to extend the study to other types of clouds. On the contrary, if the focus is only on high thick clouds it has to be more clearly stated. More generally the evaluation of retrieval algorithms on a single atmospheric simulation is probably hard to support, although this does not question the algorithm itself.

**Technical corrections**

l.4: The combined use of OSSE and retrieval is a bit misleading. Although both approaches rely on the optimal estimation, retrieval aims at retrieving optimal parameters, while OSSE generally results in weather forecasts that can be compared to forecasts without the novel instrumentation included.

l.5: "Detecting" is unclear or at least too qualitative, and different than retrieving (quantitative)

l.12: *and* ice particles

l.17: what do you include in "atmospheric composition"?

l.22: "severe" is qualitative, what does it mean in terms of cloud fraction or optical thickness for instance?

l.22: more details should be given on cloud-clearing, and why clouds generally make it difficult to retrieve atmospheric composition. More generally demonstrate more convincingly that what you're proposing has never been done before.

l.24: "large sampling footprint" is unclear. Do you mean vertical resolution? Give some numbers to illustrate that resolution is too coarse, and detail what's the typical vertical resolution that you need to investigate water vapor injection from convective towers

l.26: clarify "single footprint". What quantity do you refer to for "retrieval"? If it is water vapor, make it clear because so far it's mixed with "atmospheric composition"

l.29: how can the reader know whether these DFS values are much or not?

l.34: maybe just say the "actual complexity of clouds"

l.37: rephrase this. Suggestion: "Moreover cloud IR emission, that depends on cloud microphysics (or something similar to explain why it's critical to know the details of the clouds), largely contributes to satellite...

l.38: what does this lapse rate look like?

l.46: precise "*in situ* observations" and clarify with respect to the observations used in Feng and Huang (2018)

l.47: more details about OSSE should be provided in the introduction. It's not clear why this cannot be named "sensitivity study based on synthetic observations from NWP model outputs". OSSE generally refers to NWP to estimate the potential impact of a new type of observations on the quality of the forecast. It implies to run a NWP model with data assimilation.

l.56: why studying a tropical cyclone rather than a convective tower?

l.60: again this really looks like a retrieval rather than an OSSE.

l.70: how do the 13 hPa relate to the altitude targeted in the paper?

l.71: should any temporal spinup or exclusion of outer points of the domain be mentioned here?

l.76: can you precise whether you handle partial cloud fractions or only cloud fractions of 0 and 1 at each level

l.79: are these liquid clouds accounted for in the radiative transfer simulations? The information should be put here

l.81:  generates

l.80: is this snapshot used in Figure 1 only, or it's the only one used in the present study? What's the limitation of building and evaluating a retrieval algorithm on a single atmospheric situation?

l.81: what do you mean by "suitable"?

l.83: it is surprising to look at brightness temperature to see the cloud distribution. Looking at condensed water (liquid + ice) paths or cloud fraction would be more standard. Unless you're interested in the vertical distribution here, in which case this should be made clear

l.85: overshooting should be rigorously defined from a cloud perspective

l.85: explain the physics behind this brightness temperature difference (BTD), accounting for temperature profile in the UTLS and water vapor and cloud optical properties at these channels. What kind of BTD are expected?

l.87: "this" criterion is not clear because it was loosely defined

Figure 1: remove "of upwelling radiances". Inte~r~grated

l.103: clarify what are the pressure levels of MODTRAN, and explain how interpolation from GEM to MODTRAN is performed

l.104: "cloud information" is unclear. Clearly state "optical properties". Clarify how liquid clouds are treated when present in GEM.

l.106: what input is taken from GEM for clouds? Ice water content, liquid water content, number concentrations, effective radius? How are clouds specified in MODTRAN? Via optical thickness, single scattering properties? Please clarify how GEM information (vertical profiles) is converted into MOTRAN-required information

l.109: ok for liquid clouds, but the information probably comes too late (see above). Can you then give the range of optical thicknesses obtained for ice clouds to support this choice

l.111: reference for AIRS should appear earlier on

l.113: $cm^{-1}$ should not be italic. Holds elsewhere, including figures captions.

l.113: what's the spectral resolution of AIRS? Is the resolution of MODTRAN sufficient to simulate AIRS radiances?

l.114: remove "reference level"

l.115: does random mean "normal distribution of noise?

l.115: would you have any reference to support the uncorrelation between spectral channels? In particular, any spectral shift is expected to alter similarly all channels

l.119: is optical thickness conserved when effective radius is varied?

l.121: this sentence is unclear

l.123: DARDAR product should have been fully defined at the first occurrence

l.124: clarify whether this DARDAR selection has been done by the authors or in the cited paper

l.129: does cloud top correspond to the topmost layer of DARDAR? In the legend of Figure 2 it corresponds to 100 hPa. Make sure that this is consistent

l.132: how is roughness quantified? If solid columns are chose, is the sensitivity to habit investigated later on, as suggested in the introduction?

l.134: does this average conserve the whole mass of ice? Is the average performed over a single profile (to make it vertically homogeneous) or across all profiles to have comparable profiles?

l.135: clarify whether Re is a profile here

l.138: all this paragraph is unclear, it's not clear how the profiles are built to compute the mean and standard deviation of brightness temperatures. In particular because the vertical extent of various clouds is different.

l.144: what are the ranges of IWC and effective radius resulting in the observed variability?

l.147: is there a good reason to compute the spectra over the far-infrared, if not used further? This kind of information could be kept for discussion maybe. Note also that for FORUM you can now cite Palchetti et al. (2020).

l.147: $cm^{-1}$ should not be italic

l.154: why not to conserve IWP in this slab assumption? Unit should be g $m^{-3}$. Is this IWC commonly observed? What about the value of Re in this slab?

l.157: not clear what these residuals mean. I'd say that they highlight the uncertainties due to the slab-cloud assumption.

l.161: could this tilted bias be mostly related to the penetration depth of radiation, or equivalently to the absorption coefficient of ice (actual emission comes from different effective heights depending on wavenumber)

l.161: is this statement about RMSE and bias general? In the case you'd just have a spectrally flat bias, bias and RMSE would probably be the same, and would both vanish with debiasing.

Figure 3: probably $R(Re_0, IWC)$ for the second line

l.170: F/x should probably be derivatives

l.176: clarify whether K is computed at $x_0$, or whether it's computed iteratively and the final one is that on the best estimate

l.178: isn't Eqn. 5.16 of Rodgers (2000) more appropriate?

l.183: again, some convincing justification of this uncorrelation would be appreciated

Figure 4: *at* a spectral resolution

l.187-188: probably redundant

l.202: what about the *a priori* on IWC and Re?

l.203: it would be interesting to see how the final estimates of Re and IWC differ from the DARDAR *a priori*

l.205: I don't understand the tentative justification for using log(IWC) in the state vector

l.210: not clear why you perturb the profiles in this way. Could you simply propagate the observation error as it is done for the radiances? In this case the retrieval might be optimistic, but the associated error would reflect the IWC measurement uncertainty. It depends whether you're looking at retrieved values or final uncertainties.

l.212: microns

l.229: what's the rationale for this choice of synthetic observation? Does it correspond to a particular satellite? What if cloud conditions have completely changed? How to compute the observation vector for this? Is unity matrix chosen?

l.235: why not using $y_{atm}$ as the *a priori* instead, and increasing *a priori* error? This holds for DARDAR observations as well

l.237: what variables are you focusing on to assess the performance? All of them or just water vapor?

l.242: this experiment is surprising, you just expect to converge to DARDAR and atmospheric observations don't you?

l.244: this short paragraph is surprising. What's the use? Why does it appear now and not earlier? For instance, information content has not been defined so far

l.247: water vapor "perturbations" is too vague. Where? At which temporal frequency?

l.249: why focusing on slab-cloud assumption only here?

l.257: I guess $t_{cold}$ should have been defined in the previous paragraph as it is used in Fig. 4.

l.257: $CO_2$ not italic

l.264: shouldn't the index 'i+1' be removed?

l.266: not clear whether the cases remain the same if you now only keep radiance observations. Do you remove other observations only when computing A, but do the calculation at the retrieval point estimated with all observations? To be clarified

l.269: "As expected" is not obvious because Fig. 4 suggests that the slab-cloud assumption does not really impact the information content

l.275: maybe say "highlights the strong sensitivity"

l.277: reference to Fig. 4b is not obvious. Do you mean reference to slab-cloud error?

l.278: here the DFS for IWC with only radiances is considered. Does it prove that there would still be an added value when considering direct observations of IWC through DARDAR?

l.279: can you elaborate on this high sensitivity to low IWC on a physical basis?

l.281: that DFS for Re is surprisingly low, even though Figs. 3a and 3b suggest it should be less than DFS for IWC

l.284: maybe gather all information relative to retrieval performance here (see l.237)

l.288: you probably don't need this recap if the cases have been clearly defined before (and ideally defined when needed, that is not too early)

l.292: "there are some DFS values" is awkwardly

l.301: the patterns are not obvious. Do you refer to differences between the center of the cyclone and the points in the South-East corner of the domain, or within the cyclone? In general, Figure 6 is not as informative as the others in the way it is used in the text.

l.305: check syntax

l.305: where does this 13.5% come from?

Figure 7: why not showing truth and retrieval for IWC? Not clear what 7c and 7f show: true or retrieved profiles?

l.311: "broad"

l.311: moistening is not so clear in Fig. 7b for Case 1

l.323: "better resemblance of Truth" is awkward. Improvement from Case 1 to 3 is not obvious in Fig. 6 for water vapor profile

Figure 5: RMSE missing in title of 5f. Not clear why $y_{IWC}$ is biased. Maybe clarify how it is built, as I was expected some random white noise around the truth.

l.335: This last sentence is not clear

l.338: what is the uncertainty range? Clarifying differences between using observations to refine the *a priori* or to complete radiance observations would help.

l.340-341: that's not easy to follow. Consider reexplaining how $y_{atm}$ error is set and whether various errors are tested

l.343: "vertical" variability?

l.354: "synergistic". Also check "synergetic" earlier on

l.372: "best resemble" is awkward

l.377: consider removing the citation to an apparently non-existing paper

Table 1: in caption, 4 or 5 cases? What does 20 refer to in the DFS? Why are the DFS values similar across cases? Why are DFS values shown here and not in Table 2?

Table 2: in caption, 4 or 5 cases?

**References**

Arnold Jr, C. P., & Dey, C. H. (1986). Observing-systems simulation experiments: Past, present, and future. *Bulletin of the American Meteorological Society*, *67*(6), 687-695.

Palchetti, L., Brindley, H., Bantges, R., Buehler, S. A., Camy-Peyret, C., Carli, B., ... & Serio, C. (2020). unique far-infrared satellite observations to better understand how Earth radiates energy to space. *Bulletin of the American Meteorological Society*, *1*(aop), 1-52.

---

## Community Comment (CC1)

DETAILED COMMENTS for AMT Manuscript No. AMT-2020-518 "An observing system simulation experiment (OSSE)-based assessment of the retrieval of above-cloud temperature and water vapor using hyperspectral infrared sounder" by J. Feng et. al

The paper is an extension of the 2018 JOAT paper "Cloud-Assisted Retrieval of Lower-Stratospheric Water Vapor from Nadir-View Satellite Measurements," by J. Feng and Y. Huang, where the authors used a physically based retrieval that when compared to in-situ data, performed better than the AIRS L2 operational cloud clearing and subsequent retrieval above deep convective clouds. The AIRS L2 algorithm is more designed for tropospheric retrievals, and some of the inaccuracies could be attributed to that. In addition the cloud clearing process degrades the spatial resolution of the AIRS observations (15 km) to a 45 km footprint. The current paper is an significant extension of the previous paper. Measurements of UTLS water vapor is important for climate studies, and this paper is an important step towards understanding how to improve the retrievals above convective storms, for subsequent use by the scientific community.

I find the paper well written, and would be a very welcome addition to the literature. Though I understand the paper is purely a simulation exercise, the fact that we currently enjoy almost 20 continuous years of high quality, low noise hyperspectral radiance measurements, means the paper would benefit from some corrections and/or improved explanations, and a discussion of how the consequences/limitations of realistic instrument parameters would affect your simulated results.

Primarily the addition of the T/WV profiles a few hours "after" the event to the observation vector is puzzling. I'm not an expert on dynamics, so could the authors explain why 400 minutes (7 hours) were chosen? Is this some (intensification?) timescale associated with tropical cyclonic activity? Furthermore the authors have simulated a tropical cyclone, which lasts for a few days and hence is well tracked with NWP fields populated by data assimilation. I can't see how this would help for a mesoscale outburst over a land mass, which would have timescales of hours?

Figures 3(a) 3(b) and Figure 4 are misleading. Current Infrared sounders have detectors between 650 cm-1 to 2780 cm-1, so adding far infrared channels to your OSSE could be unrealistically "helping" the retrieval. Can you remove these channels and comment how the retrieval is affected, for example in terms of degrees of freedom?

Furthermore, the spectrally constant 0.3 K noise estimate is extremely conservative by today's standards. For a 250 K observation, the noise in the high altitude 15 um temperature sounding channels is close to 0.6 K; for the 200 K simulated observations, the noise would be larger than 1 K (!). Similarly in the WV sounding region (1300-1700 cm-1) the noise would increase from about 0.2 K (close to what you use in your OSSE) to about 0.6 K. Again could you test and comment how this affects your retrieval performance?

All these points should combine to lower the Degrees of Freedom that you retrieve. For the reasons above, the numbers you quote (3 for T(z), almost 3 for IWC) seem very optimistic for what is being retrieved from 13 km to 80 km, where the typical sounder channels rapidly run out of steam.

Another point that should be mentioned is the high altitude channels have very narrow doppler linewidths. For example in the 1500 cm-1 WV region, a LBL code would show lines with widths on the order of 0.05 cm-1; the linewidths would decrease even further as you get deeper into LW IR or FarIR (eg the temperature sounding channels at 15 um). So 0.1 cm-1 from MODTran is not sufficient, even if you then convolve over an AIRS SRF which is typically 0.5-2 cm-1 wide.

Finally, you use CloudSat cloud profiles in your "observation" state vector (Page 9, line 205); but the satellite with that instrument on board has moved away from the A-Train and you no longer have co-located measurements. Besides it was purely a nadir looking instrument, and hence would more likely than not miss the DCC that AIRS regularly see. In your conclusions (or somewhere else) you should mention these facts.

Below are some additional points I have come across :

1. Abstract : Since this is an OSSE, so all your work is with simulated radiances. Hence in the second line of the abstract, please write "non-negligible impact on simulated TOA infrared radiances". By the way you need to define what TOA means.

2. Page 1, line 20-23 : as stated above AIRS L2 does not focus on UT/LS water vapor, though the process of cloud clearing should work best in the non-homogeneous conditions that occur when there are DCC mixed in with overshooting clouds. Perhaps you could introduce the names (and acronyms) of current sounders in line 23, around where you mention Susskind's paper etc.

3. Page 2, line 29 : Please also include Irion, F. W., Kahn, B. H., Schreier, M. M., Fetzer, E. J., Fishbein, E., Fu, D., Kalmus, P., Wilson, R. C., Wong, S., and Yue, Q.: Single-footprint retrievals of temperature, water vapor and cloud properties from AIRS, Atmos. Meas. Tech., 11, 971–995, https://doi.org/10.5194/amt-11-971-2018, 2018

4. Page 3, line 85-90 : Would you consider showing a MODIS image, and an AIRS BT1231 cm-1 image for this May 16, 2015 storm? I can see AIRS granule 41 for that date would work (see also coincident MODIS 4.05 AM image). This maybe a good check of the realism of the simulations (though yours were performed for about 7 AM of that day). I did a quick check and there were 242 AIRS observations with BT1231 colder than 200 K.

5. Page 6, Line 145-146 and Figure 3 : you may want to point out eg 667 cm-1 and 2300-2360 cm-1 are very high altitude temperature sounding channels, whose weighting functions peak way above the tropopause (and hence the clouds you put in); similarly 1000 cm-1 region is the ozone sounding region, and O3 weighting functions peak in the stratosphere.

6. Page 6, Line 145-146 and Figure 3 : so it looks like other than the difference in magnitude, the effective radius and IWC jacobians are very similar, so it is quite difficult to unscramble them, other than constraining the effective radius in your retrieval. For completeness, you should discuss what the cloud top/cloud bottom jacobians look like.

7. Page 10, Figure 4 panels (a),(c) refer to my earlier comment, namely current sounders have detectors between 640-2700 cm-1, and the noise levels at 640-720 cm-1 are far larger than you use.

8. Page 10, Figure 4(b) is very interesting. Very often AIRS DCC observations at 1231 cm-1 are far colder than NWP tropopause temperatures, so that kink at 80 mb is required. Are there any consequences to the stability of the atmosphere and the tops of overshooting DCC, since your cold profile snaps back to the average stratospheric profile very rapidly? (by about 75 mb!!!).

9. Page 11, Line 225 : Please give a reference for the mapping from one vertical grid to another (eg the remote sensing book by Rodgers 2000, or some of the TES retrieval algorithm papers by eg Worden or Kuwalik. Irion's 2018 AMT paper mentioned above should also have that).

10. page 14, line 315 : replace "board" with "broad"

11. Figure 5, page 15 : I am impressed by the DOFs and ability of your retrieval to get the overall details correct. However as mentioned above the AIRS NeDT is much larger than you use in your OSSE. Plus the AIRS channels typically has less information content for higher altitudes. So could you run a quick check of how this would impact retrieval performance?

12. Figure 6, page 16 : please check the caption, I think you mean rows 1-5, not columns 1-5; if so there are other sentences in the caption that need fixing.

13. Could you mention what CO2 and CH4 ppm value you used for your retrieval?

14. You have not explicitly stated which trace gases you retrieved (ozone? CH4?), if any. This goes back to assuming you are using all channels shown in Figures 3,4. Assuming you did use all channels but did not fit eg CH4 and O3 and CO, perhaps you could get improved results if you removed the spectral regions that those gases cover?

---

## Author Comment (AC2)

**Response to Quentin-Libois**

This paper is a follow-up of a previous study (Feng and Huang, 2018) that investigated the feasibility to retrieve stratospheric water vapor profiles above thick convective clouds using infrared hyperspectral observations. The present study explores in details the performances of various retrieval strategies (based on the optimal estimation method) and in particular the added value of using ancillary observations (ice cloud properties from the DARDAR synergistic product and atmospheric profiles in the vicinity of the scene) to retrieve water vapor, temperature, and ice water content profiles. Including vertical information on the distribution of ice near the top of the cloud greatly improves the retrievals compared to the slab-cloud hypothesis where an optically thick cloud is located at a single level in the retrieval algorithm, mostly when the top of the cloud exhibits low ice water contents. Using additional atmospheric information also improves the retrieval compared to using climatological a priori values.

The various retrieval strategies are generally well presented and detailed. The paper is well written and the figures appropriate. The manuscript, rather technical, perfectly fits for AMT journal. However the abstract lacks information and the introduction is probably too short to highlight the originality of the work compared to previous studies. A particular critical point is the use of the term "OSSE" to describe something that is probably more a standard retrieval, although tested on atmospheric fields originating from an NWP model. The multiplicity and relative complexity of the different cases described makes it sometimes difficult to follow the optimal estimation framework, but local rephrasing can certainly improve the readability. The results should probably put in a broader context to help the reader figure out whether the improvements for the various retrievals mentioned are significant, and how they compare to existing methods. This paper deserves being published in AMT, but the authors are encouraged to make their best to make the reading as easy as possible. Various suggestions are made below towards this.

**Thank you, Quentin Libois, for your detailed and thoughtful comments. Following your comment, we have greatly expanded the abstract and introduction, added necessary description about the forward model, and expanded the discussion on cloud-induced uncertainties.**

Specific comments

1) The abstract is not an abstract so far. An abstract should indeed be a very short summary (very similar to the conclusion actually) of the work. It could be a bit longer, emphasizing for instance that generally sounders reject cloudy pixels (at least in NWP models). It should highlight quantitative results. An abstract should be self-sufficient and provides details : what instruments are used, how much are the performances improved etc. ? The abstract needs rewriting.

**Following this comment, we re-wrote the abstract.**

2) The term "OSSE" is used, in particular in the title, to define the type of experiment performed in this study. Although NWP model outputs are used to test the retrieval algorithms, the present study is not an OSSE, in the sense that no forecasts are performed with the NWP models. In general, an OSSE implies to compare two NWP forecasts, one of those including the assimilation of observations from a supplementary instrument compared to the other. The gain in forecast skill between both forecasts can then be attributed to the additional use of this new instrument. See for instance Arnold and Dey (1986).

**We agree that the term "OSSE" is typically used to describe the weather OSSE that assesses the added value from satellite observations to operational observation systems (e.g., NWP forecast).**

**However, many studies (such as Feldman et al., 2013, Smith et al., 2020, Crow et al., 2001) also used this term to describe simulation experiments that derive observational signals from numerical weather forecast model or climate model to evaluate the information content carried by the new instruments, without the assimilation of the observations.**

**Following your suggestion, we use 'simulation experiment' instead.**

3) Some descriptions of the retrieval algorithms are not very clear, in particular when it comes to the

details of the optimal estimation method, with the definition of observation vectors, state vectors and covariance errors. In particular Case 5 experiment is not sufficiently clear. It is essential that the combination of the text and equations allows the reader to reproduce the algorithm. The same holds for the definition of DFS relative to each state variable for instance. Likewise, the way atmospheric profiles are sometimes modified for sensitivity studies is not always clear (see examples below).

**To avoid confusion, we now refer to Cases 1-4 as 'retrieval strategies', and avoid referring Case 5 as a retrieval. Necessary clarifications on atmospheric profiles for sensitivity studies are added.**

4) The study deliberately focuses on high thick clouds to probe UTLS atmospheric properties. However the title seems to be more general and does not exclude other types of clouds, including for instance low clouds or thin ice clouds. To make the paper more general it'd be useful to extend the study to other types of clouds. On the contrary, if the focus is only on high thick clouds it has to be more clearly stated. More generally the evaluation of retrieval algorithms on a single atmospheric simulation is probably hard to support, although this does not question the algorithm itself.

**A focus on high, thick clouds is now stated in the abstract, as well as the introduction. We do not expand it to cloudy-sky cases in general, considering: 1) a general application using a similar single-footprint OE framework has been discussed in Irian et al., 2018 and DeSouza-Machado et al., 2018, and 2) the way our radiative transfer model is configured is to neglect the contribution from lower atmosphere.**

**Technical corrections**

l.4: The combined use of OSSE and retrieval is a bit misleading. Although both approaches rely on the optimal estimation, retrieval aims at retrieving optimal parameters, while OSSE generally results in weather forecasts that can be compared to forecasts without the novel instrumentation included.
**See response 2.**

[l.5: "Detecting" is unclear or at least too qualitative, and different than retrieving (quantitative)

l.12: *and* ice particles]
**Abstract has been re-wrote.**

[l.17: what do you include in "atmospheric composition"?
l.22: "severe" is qualitative, what does it mean in terms of cloud fraction or optical thickness for instance?
No clear-sky portion in nine (3x3) FOVs (Overcast cloud cover)
l.22: more details should be given on cloud-clearing, and why clouds generally make it difficult to retrieve atmospheric composition. More generally demonstrate more convincingly that what you're proposing has never been done before.
Done
l.24: "large sampling footprint" is unclear. Do you mean vertical resolution? Give some numbers to illustrate that resolution is too coarse, and detail what's the typical vertical resolution that you need to investigate water vapor injection from convective towers
Horizontal resolution. Done.
]
**We re-wrote this paragraph to describe the current in-situ and satellite observations in UTLS water vapor. See L19-43.**

l.26: clarify "single footprint". What quantity do you refer to for "retrieval"? If it is water vapor, make it clear because so far it's mixed with "atmospheric composition"
**The cloud-clearing algorithm uses adjacent nine footprints while the references given in this paragraph uses only one footprint.**

l.29: how can the reader know whether these DFS values are much or not?

**We re-phrased this sentence. However, whether DFS values are much depends on readers' study of interest, which is the purpose of giving DFS values.**

l.34: maybe just say the "actual complexity of clouds"
**We re-phrased this sentence**

l.37: rephrase this. Suggestion: "Moreover cloud IR emission, that depends on cloud microphysics (or something similar to explain why it's critical to know the details of the clouds), largely contributes to satellite...
**We re-phrased this sentence.**

l.38: what does this lapse rate look like?
**We re-phrased this sentence. Deep convective clouds are typically associated with a cold cloud top, due to radiative and dynamical cooling. It is however warmer within the cloud as a result of latent heat release and radiative heating. Therefore, temperature decreases dramatically near the cloud top.**

l.46: precise "*in situ* observations" and clarify with respect to the observations used in Feng and Huang (2018)
**To answer your question, the observation used in Feng and Huang 2018 is from Anderson et al., 2012, which is also cited here. Using the in-situ observations carried by aircraft campaigns some limitations in validations of the retrieval. The 'collocation' is never exact. For safety/technical concerns, aircraft samples around a convective system, instead of flying above them. There will always be a certain time/space mismatch.**

l.47: more details about OSSE should be provided in the introduction. It's not clear why this cannot be named "sensitivity study based on synthetic observations from NWP model outputs". OSSE generally refers to NWP to estimate the potential impact of a new type of observations on the quality of the forecast. It implies to run a NWP model with data assimilation.
**See response 2.**

l.56: why studying a tropical cyclone rather than a convective tower?

**One purpose of developing the retrieval approach is to detect stratosphere humidification. Tropical cyclone events can create horizontally extensive cloud cover with overshooting tops, which the retrieval method described in the manuscript can be effectively applied and then contrast the overshooting tops with the 'background' to show up the effect of overshooting. For mid-latitude storms,**

**We stated the focus on high, thick clouds. Compared to a single convective tower in the mid-latitude, a single tropical cyclone event can produce large high, thick cloud cover for several hundreds of kilometers. This is also relevant to your later question, 'L80', so that a single event produces relatively large variability in temperature, water vapor, and cloud fields.**

**This event particular event is selected because of the availability of collocated aircraft campaigns (ATTERX) and AIRS overpass, which can help further validation of the retrieval algorithm. However, we chose not to mention it in the paper because such validation studies are not performed here.**

l.60: again this really looks like a retrieval rather than an OSSE.
**See response 2.**

l.70: how do the 13 hPa relate to the altitude targeted in the paper?
**29.1 km.**

l.71: should any temporal spinup or exclusion of outer points of the domain be mentioned here?
**Done.**

l.76: can you precise whether you handle partial cloud fractions or only cloud fractions of 0 and 1 at each level
**The NWP simulates partial cloud fraction and we consider cloud fraction when generating Figure 1(a). However, we find it is not necessary to consider cloud fraction as in this time instances (Figure 1 (a)), most deep convective clouds have a cloud fraction of 1. See 140, the definition of overshooting deep convective clouds and Figure 1(a).**

l.79: are these liquid clouds accounted for in the radiative transfer simulations? The information should be put here

**No. We add clarification in the text.**

l.81:  generates
**Done.**

l.80: is this snapshot used in Figure 1 only, or it's the only one used in the present study? What's the limitation of building and evaluating a retrieval algorithm on a single atmospheric situation?

**It is the only one used in the present study. For evaluating high, thick clouds only, we think using this snapshot is sufficient, as it provides enough samples and generates different combinations of temperature, water vapors profiles. By estimating the standard deviation for every 10 minutes simulation throughout 24 hours, we find this snapshot is representative of the variability of this event.**

**As tropical cyclone systems are unique in generating an outflow that affects a large spatial domain, other overshooting convections in mid-latitude storms can be different. We will incorporate mid-latitudes storms to generate a similar simulation study for the application of the method on convective storms other than tropical cyclones and evaluating more detaily on the averaging kernel and sampling footprints.**

l.81: what do you mean by "suitable"?
**See previous response. Descriptions added.**

l.83: it is surprising to look at brightness temperature to see the cloud distribution. Looking at condensed water (liquid + ice) paths or cloud fraction would be more standard. Unless you're interested in the vertical distribution here, in which case this should be made clear.

**I agree that the water path is typically used to indicate cloud distribution. However, the purpose of showing Figure 1 is to illustrate a hypothetical infrared satellite image that is very commonly used for infering the distribution of clouds, especially overshooting and deep convective clouds, from satellite observations.**

l.85: overshooting should be rigorously defined from a cloud perspective

**In the original manuscript, we do look into precipitation, cloud fraction, and the vertical extent to identify overshooting from a cloud perspective. It was explicitly stated.**

**Not all OT-DCCs pass the BT criterion, because the simulated cloud fields near the edge of DCCs can have very low IWC, even though they meet all the criteria described above. The BT criterion is designed to minimize the false positive rate. We chose not to justify the BT criterion here because it is quantitatively examined by the cited previous studies and by an accompanying paper.**

l.85: explain the physics behind this brightness temperature difference (BTD), accounting for temperature profile in the UTLS and water vapor and cloud optical properties at these channels. What kind of BTD are expected?
**Done.**

l.87: "this" criterion is not clear because it was loosely defined.
**Done.**

Figure 1: remove "of upwelling radiances".
**Done.**

l.103: clarify what are the pressure levels of MODTRAN, and explain how interpolation from GEM to MODTRAN is performed
**Done. Section 2.2 has been expanded. Although we do not perform interpolation in this simulation experiment. Layers from 13.5 hPa to TOA (0.1hPa) are added to radiative transfer model.**

l.104: "cloud information" is unclear. Clearly state "optical properties". Clarify how liquid clouds are treated when present in GEM.
**Section 2.2 has been expanded.**

l.106: what input is taken from GEM for clouds? Ice water content, liquid water content, number concentrations, effective radius? How are clouds specified in MODTRAN? Via optical thickness, single scattering properties? Please clarify how GEM information (vertical profiles) is converted into MOTRAN-required information

**IWC at every vertical level is inputted. Particle size distribution, effective radius, roughness, habit are used to build a look-up table for the extinction coefficients, single-scattering albedo, and asymmetry factors, defined over ice water content.**

l.109: ok for liquid clouds, but the information probably comes too late (see above). Can you then give the range of optical thicknesses obtained for ice clouds to support this choice
**Optical thickness of ice is greater than 100 in a typical DCC sample. Explanation added to Section 2.2**

l.111: reference for AIRS should appear earlier on
**Done.**

l.113: cm$^{-1}$ should not be italic. Holds elsewhere, including figures captions.
**Done.**

l.113: what's the spectral resolution of AIRS? Is the resolution of MODTRAN sufficient to simulate AIRS radiances?
**Yes. The Full Width at Half Maximum of the AIRS spectral response function is around 0.4 to 1 cm^-1. Berk et al., 2017 (Figure 8) validated the line-by-line algorithm of MODTRAN 6 with AIRS spectral response function. It is more convenient to use MODTRAN for calculating cloudy radiances as it includes DISORT. It is also faster than the traditional LBLRTM.**

l.114: remove "reference level"
**Done.**

l.115: does random mean "normal distribution of noise?
**Yes.**

l.115: would you have any reference to support the uncorrelation between spectral channels? In particular, any spectral shift is expected to alter similarly all channels.
**Correlation between spectral channels can be a concern; however, we do not find references/data sources that quantitatively document how spectral channels are correlated. Therefore, synthetic noise is assumed to be uncorrelated and the Se error covariance matrix is assumed to be diagonal.**

l.119: is optical thickness conserved when effective radius is varied?
**No. Optical thickness is not conserved. See response to 'l.106'. The radiative transfer model is driven by IWC at every level and ice optical properties over IWC. The ice optical properties over IWC are strongly dependent on effective radius.**

l.121: this sentence is unclear
**Rephrased.**

l.123: DARDAR product should have been fully defined at the first occurrence
**Done. See introduction.**

l.124: clarify whether this DARDAR selection has been done by the authors or in the cited paper
**Done.**

l.129: does cloud top correspond to the topmost layer of DARDAR? In the legend of Figure 2 it corresponds to 100 hPa. Make sure that this is consistent
**Rephrased.**

l.132: how is roughness quantified? If solid columns are chose, is the sensitivity to habit investigated later on, as suggested in the introduction?

**1) roughness has a negligible impact on the spectral range of interest, as it mainly affects the scattering**

angle.

**2) Crystal habit indeed has some impacts and can be important. However, we lack the habit fraction dataset, which makes it difficult to assess the impact. In the updated manuscript, we chose to include the radiance uncertainties due to the neglection of habits other than solid columns in the forward model error.**

l.134: does this average conserve the whole mass of ice? Is the average performed over a single profile (to make it vertically homogeneous) or across all profiles to have comparable profiles?
**mass of ice is always converged because IWC and effective radius is both reported by DARDAR-Cloud product and we use them independently. Average is performed across all profiles.**

l.135: clarify whether Re is a profile here
**After the revision, Re is not a profile in the retrieval process but it is a profile for generating the synthetic observation. We now add the radiance uncertainties caused by neglecting the vertical variation in Re to the forward model error.**

l.138: all this paragraph is unclear, it's not clear how the profiles are built to compute the mean and standard deviation of brightness temperatures. In particular because the vertical extent of various clouds is different.
**After revision, we clarify the t,q profiles used, and also add more description of how effective radius affects the radiance spectral in the radiative transfer calculations. Also see previous response that the description of radiative transfer calculation has been expanded.**

l.144: what are the ranges of IWC and effective radius resulting in the observed variability?
**See revised Figure 2.**

l.147: is there a good reason to compute the spectra over the far-infrared, if not used further? This kind of information could be kept for discussion maybe.
**The far-infrared spectra is kept for readers' interests. We kept this spectral range in the plot. To avoid confusion, the spectral range used in the retrieval is marked in Figure 4.**

Note also that for FORUM you can now cite Palchetti et al. (2020).
**Done.**

l.147: $cm^{-1}$ should not be italic
**Done. For other units as well.**

l.154: why not to conserve IWP in this slab assumption? Unit should be $g\ m^{-3}$. Is this IWC commonly observed? What about the value of Re in this slab?

**1) Conserving IWP in the slab is a good idea. Conserving the IWP in such a slab can have IWC around 15 g/m^3 (mean value in the DARDAR-cloud dataset for overshooting DCCs). This value is much higher than reality. We choose 1.5 g/m^3 to match the upper limit we found around 14 km for DCCs. The choice of IWC actually does not matter much in our study of interest, because any value greater than 0.2 g/m^3 for 500 m can lead to saturation (optical depth of 10).**
**2) Yes.**
**3) Re is 34 micros**

l.157: not clear what these residuals mean. I'd say that they highlight the uncertainties due to the slab-cloud assumption.
**Done.**

l.161: could this tilted bias be mostly related to the penetration depth of radiation, or equivalently to the absorption coefficient of ice (actual emission comes from different effective heights depending on wavenumber)
**Yes, exactly. The dependence of extinction coefficients on wavenumbers determines the penetration depth of radiation at different wavenumbers.**

l.161: is this statement about RMSE and bias general? In the case you'd just have a spectrally flat bias, bias and RMSE would probably be the same, and would both vanish with debiasing.

**It is not flat, because altering IWC and Re affects the optical depth and hence thermal emission, which affects the overall brightness temperature.**

Figure 3: probably R(Re$_0$, IWC) for the second line
**Done.**

l.170: F/x should probably be derivatives
**Done.**

l.176: clarify whether K is computed at x$_0$, or whether it's computed iteratively and the final one is that on the best estimate
**Done. It is iteratively computed. Although updating K (or not), except at the beginning two time steps, does not make much difference to the retrieved values.**

l.178: isn't Eqn. 5.16 of Rodgers (2000) more appropriate?
**Done.**

l.183: again, some convincing justification of this uncorrelation would be appreciated
**Done.**

Figure 4: *at* a spectral resolution
**Done.**

l.187-188: probably redundant
**Done.**

l.202: what about the *a priori* on IWC and Re?
**IWC uses the same *a priori* dataset as temperature and water vapor. Re is based on the DARDAR-Cloud Datasets. There are no correlation between the two in the covariance matrix.**

l.203: it would be interesting to see how the final estimates of Re and IWC differ from the DARDAR *a priori*
**Results for Re are added to Table 2 and IWC profiles are added to Figure. We did not show how final estimates of IWC compared to y_{iwc} because it is hard to visualize the difference.**

l.205: I don't understand the tentative justification for using log(IWC) in the state vector
**Mainly to avoid getting negative values and also because DARDAR outputs their posteriori uncertainty in log scale. Actually, we also run retrieval tests using IWC without log scale by adding a positive constraints on IWC. The final estimates and DFS do not differ much from what you see in this manuscript.**

l.210: not clear why you perturb the profiles in this way. Could you simply propagate the observation error as it is done for the radiances? In this case the retrieval might be optimistic, but the associated error would reflect the IWC measurement uncertainty. It depends whether you're looking at retrieved values or final uncertainties.

**There are DFS in IWC and Re so we would like to update them. In practice, using it as the first guess for iwc or as 'y_{iwc}' gives similar results (if not identical). We choose to formulate the retrieval this way because we want to use the climatology (a priori) in the x0 and Sa and use observations only in y, to distinguish the source of error.**

l.212: microns
**Done.**

l.229: what's the rationale for this choice of synthetic observation? Does it correspond to a particular satellite? What if cloud conditions have completely changed? How to compute the observation vector for this? Is unity matrix chosen?
**This choice is arbitrary.**

l.235: why not using $y_{atm}$ as the *a priori* instead, and increasing *a priori* error? This holds for DARDAR observations as well

**See previous response and the Turner and Blumberg (2018). In this study, because we do not have information in lower-altitudes from infrared radiances, this framework is especially helpful. The state vector in lower-altitudes sticks to the y_atm, (i.e., reanalysis). We find that use 'y_{atm}' as the observation constraint, rather than *a priori*, greatly improves the convergence when using real observations. There might be other technics to improve the convergence, but I find this framework best describes the source of uncertainty.**

l.237: what variables are you focusing on to assess the performance? All of them or just water vapor?
**See Table 2.**

l.242: this experiment is surprising, you just expect to converge to DARDAR and atmospheric observations don't you?
**Yes. In the revision, we avoid referring to case 5 as a retrieval. It is the posterior estimation of the state vector without infrared radiances. We conduct it so that readers can statistically compare it to Case 4 and infer the improvement from infrared radiances.**

l.244: this short paragraph is surprising. What's the use? Why does it appear now and not earlier?
For instance, information content has not been defined so far
**Deleted.**

l.247: water vapor "perturbations" is too vague. Where? At which temporal frequency?
**Rephrased it.**

l.249: why focusing on slab-cloud assumption only here?
**Rephrased it.**

l.257: I guess $t_{cold}$ should have been defined in the previous paragraph as it is used in Fig. 4.
**Done.**

l.257: $CO_2$ not italic

**Done.**

l.264: shouldn't the index 'i+1' be removed?
**Done.**

l.266: not clear whether the cases remain the same if you now only keep radiance observations. Do you remove other observations only when computing A, but do the calculation at the retrieval point estimated with all observations? To be clarified
**Clarified in Section 3. Yes, I only remove other observation only when computing A, because I want to show the DFS from radiance observation alone.**

l.269: "As expected" is not obvious because Fig. 4 suggests that the slab-cloud assumption does not really impact the information content
**Yes, you are right. We rephrased it.**

l.275: maybe say "highlights the strong sensitivity"
**Done.**

l.277: reference to Fig. 4b is not obvious. Do you mean reference to slab-cloud error?
**Yes. Clarified.**

l.278: here the DFS for IWC with only radiances is considered. Does it prove that there would still be an added value when considering direct observations of IWC through DARDAR?
**Yes, exactly.**

l.279: can you elaborate on this high sensitivity to low IWC on a physical basis?
**Done.**

l.281: that DFS for Re is surprisingly low, even though Figs. 3a and 3b suggest it should be less than DFS for IWC
**Thank you for pointing it out. You are right. There was a mistake in calculating kernels. Corrected in the revision.**

l.288: you probably don't need this recap if the cases have been clearly defined before (and ideally defined when needed, that is not too early)
**Deleted.**

l.292: "there are some DFS values" is awkwardly
**Rephrased.**

l.301: the patterns are not obvious. Do you refer to differences between the center of the cyclone and the points in the South-East corner of the domain, or within the cyclone? In general, Figure 6 is not as informative as the others in the way it is used in the text.
**Rephrased.**

l.305: check syntax
**Done.**

l.305: where does this 13.5% come from?
**exp(-2), the transmittance through a layer with optical depth of 2.**

Figure 7: why not showing truth and retrieval for IWC? Not clear what 7c and 7f show: true or retrieved profiles?

**Clarified in Figure 7. Because truth, retrieval, and y_iwc are visually similar. It is hard to tell which is which. We added them to Figure 7.**

l.311: "broad"

**Done.**

l.311: moistening is not so clear in Fig. 7b for Case 1
**Yes, because of the smearing effect as explained. You can still see over a higher mixing ratio over a 40 hPa vertical range.**

l.323: "better resemblance of Truth" is awkward.
**Done.**
Improvement from Case 1 to 3 is not obvious in Fig. 6 for water vapor profile
**Yes, still because of smearing effect. Including y_atm increases DFS, leading to narrower vertical averaging kernel, and therefore a narrower peak in Case 4.**

Figure 5: RMSE missing in title of 5f. Not clear why $y_{IWC}$ is biased. Maybe clarify how it is built, as I was expected some random white noise around the truth.
**Thank you for pointing it out. We found that the number generated by a Matlab function was not really random. We have overcome this issue.**

l.335: This last sentence is not clear
**Rephrased.**

l.338: what is the uncertainty range? Clarifying differences between using observations to refine the a priori or to complete radiance observations would help.
**We work on the uncertainty range more on an accompanying paper. Please check Figure A2, https://doi.org/10.5194/acp-2021-154**

l.340-341: that's not easy to follow. Consider reexplaining how $y_{atm}$ error is set and whether various errors are tested
Done.

l.343: "vertical" variability?
**Done.**

l.354: "synergistic". Also check "synergetic" earlier on
**Done.**

l.372: "best resemble" is awkward
**Re-phrased.**

l.377: consider removing the citation to an apparently non-existing paper
**Removed.**

Table 1: in caption, 4 or 5 cases? What does 20 refer to in the DFS? Why are the DFS values similar across cases? Why are DFS values shown here and not in Table 2?

Table 2: in caption, 4 or 5 cases?

1) **Clarified. We now state four 'retrieval' cases, because case 5 is not a retrieval. 2) '20' refers to the number of model layers used to calculate DFS in t and q. 3) DFS is similar because uncertainties caused by the slab-cloud assumption is small. 4) Because it is hard to merge Table 1 and 2 because the limit of page size.**

---

## Author Response (AR1)

**Point-to-Point Response AMT-2020-518**

We would thank Dr. Quentin Libois, and Dr. Sergio DeSouza-Machado, the anonymous reviewer, and Dr. Jun Wang for their time and contribution to the review of this paper. Following their comments, several major changes have been made:

- **1.** Expanded the abstract and the introduction.
- 2. Clarified cloud inputs to the radiative transfer model and the noise level used in the simulation experiment.
- 3. Improved AIRS channel selection.
- 4. Discussed radiance uncertainties caused by column-to-column (horizontal) and layer-to-layer (vertical) variations in effective radius. Added a forward model error caused by simplified cloud inputs of the radiative transfer model to the synergistic method.

Please check point-to-point responses to all comments below.

**Response to Anonymous Referee**

This paper describes a method of improving the retrieval of temperatures and ice specific humidy using hyperspectral infrared measurements with added synergetic measurements of cloud effective radius and ice water content (IWC). A cloud resolving NWP model is used to generate a scene which acts as the "truth", which is input to MODTRAN for simulating the infrared radiances. The retrieval algorithm used two assumptions. One is the slab assumption which use the simulated radiances as inputs and the other uses synergetic dataset including IWC and cloud effective radius as synthetic measurements. Improvements have been found by adding these synergetic inputs.

The paper is well written except that there are minor typos here and there. For example, Fig. 6 refers "row" as column. Line 216 use um instead of  $\mu$ m for micron. I suggest that the authors do a thorough proof-reading and eliminate these typos.

The abstract is not as informative as I would see from a high quality paper. It merely mentioned that the algorithm is able to detect the spatial distribution of temperature and humidity anomalies above convective storms. Some more quatative results should be summarized here.

Overall, I recommend the publication of this paper with the minor revision incorporated.

Thank you for the general assessment of our manuscript and for your timely review comments. Following your suggestions, we have rewritten the abstract and proofread the manuscript to clear up language mistakes and typos.

**Response to Sergio DeSouza-Machado**

The paper is an extension of the 2018 JOAT paper "Cloud-Assisted Retrieval of Lower-Stratospheric Water Vapor from Nadir-View Satellite Measurements," by J. Feng and Y. Huang, where the authors used a physically based retrieval that when compared to in-situ data, performed better than the AIRS L2 operational cloud clearing and subsequent retrieval above deep convective clouds. The AIRS L2 algorithm is more designed for tropospheric retrievals, and some of the inaccuracies could be attributed to that. In addition, the cloud clearing process degrades the spatial resolution of the AIRS observations (15 km) to a 45 km footprint. The current paper is an significant extension of the previous paper. Measurements of UTLS water vapor is important for climate studies, and this paper is an important step towards understanding how to improve the retrievals above convective storms, for subsequent use by the scientific community.

I find the paper well written, and would be a very welcome addition to the literature. Though I understand the paper is purely a simulation exercise, the fact that we currently enjoy almost 20 continuous years of high quality, low noise hyperspectral radiance measurements, means the paper would benet from some corrections and/or improved explanations, and a discussion of how the consequences/limitations of realistic instrument parameters would affect your simulated results.

**We thank Sergio DeSouza-Machado for his constructive comments. Following his suggestion, we have clarified the noise level of the instrument used in this study and improved the instrument channel selection by removing O3 and CH4 channels.**

Primarily the addition of the T/WV profiles a few hours \after" the event to the observation vector is puzzling. I'm not an expert on dynamics, so could the authors explain why 400 minutes (7 hours) were chosen? Is this some (intensification?) timescale associated with tropical cyclonic activity? *Furthermore, the authors have simulated a tropical cyclone, which lasts for a few days and hence is well tracked with NWP fields populated by data assimilation. I can't see how this would help for a mesoscale outburst over a land mass, which would have timescales of hours?*

- 1) Choosing this time step is arbitrary, simply to represent the quantitative difference between products such as reanalysis and the truth atmospheric conditions.
- 2) It is a good point considering how the addition of infrared spectra may help the data assimilation process. We have not worked on this problem so far.

Figures 3(a) 3(b) and Figure 4 are misleading. Current Infrared sounders have detectors between 650 cm-1 to 2780 cm-1, so adding far-infrared channels to your OSSE could be unrealistically \helping" the retrieval. Can you remove these channels and comment how the retrieval is affected, for example in terms of degrees of freedom?

**The far infrared channel in Figures 3 and 4 is to evaluate the effects of cloud more generally. It is not included in the retrieval. We now emphasis the AIRS-like instrument specification in the paper. To avoid confusion, we mark the AIRS spectral range used in this study in Figure 4.**

Furthermore, the spectrally constant 0.3 K noise estimate is extremely conservative by today's standards. For a 250 K observation, the noise in the high altitude 15 um temperature sounding channels is close to 0.6 K; for the 200 K simulated observations, the noise would be larger than 1 K (!). Similarly in the WV sounding region (1300-1700 cm-1) the noise would increase from about 0.2 K (close to what you use in your OSSE) to about 0.6 K. Again could you test and comment how this affects your retrieval performance? All these points should combine to lower the Degrees of Freedom that you retrieve. For the reasons above, the numbers you quote (3 for T(z), almost 3 for IWC) seem very optimistic for what is being retrieved from 13 km to 80 km, where the typical sounder channels rapidly run out of steam.

**In the OSSE, a radiometric noise from AIRS L1B is used. Therefore, the DFS shown in the original manuscript is not biased due to the noise level.**

In the original manuscript, a 0.3 K NEdT is used for plotting, just to be consistent with AIRS documentation (which is reported at a 250K reference level) and also Feng and Huang., 2018, to emphasize how the addition of cloud-induced uncertainties affects the signal detection. You are right, NEdT is indeed much higher at a cold reference temperature like you said. In the revision, we now use 0.5 K, the spectral mean value, in Figure 4.

Another point that should be mentioned is the high altitude channels have very narrow doppler linewidths. For example in the 1500 cm-1 WV region, an LBL code would show lines with widths on the order of 0.05 cm-1; the linewidths would decrease even further as you get deeper into LW IR or FarIR (eg the temperature sounding channels at 15 um). So 0.1 cm-1 from MODTran is not sufficient, even if you then convolve over an AIRS SRF which is typically 0.5-2 cm-1 wide.

A line-by-line algorithm of MODTRAN 6.0 is used. This LBL algorithm segments 100 (user-defined) subbins within the 0.1 cm-1 (for calculating the line shape of each molecular transition). The Doppler broadening width is therefore accurately accounted for. We now clarify it in the manuscript to avoid confusion. See L154. Or do you think precomputing contributions from line tails in their algorithm could be a concern?

Finally, you use CloudSat cloud profiles in your \observation" state vector (Page 9, line 205); but the satellite with that instrument on board has moved away from the A-Train and you no longer have co-located measurements. Besides it was purely a nadir looking instrument, and hence would more likely than not miss the DCC that AIRS regularly see. In your conclusions (or somewhere else) you should mention these facts.

**Thank you for pointing it out. I clarify this important fact when introducing the instruments/ products. See L111-115.**

Below are some additional points I have come across :

1. Abstract : Since this is an OSSE, so all your work is with simulated radiances. Hence in the second line of the abstract, please write \non-negligible impact on simulated TOA infrared radiances". By the way you need to define what TOA means.

**Done. The abstract is greatly expanded as well.**

2. Page 1, line 20-23 : as stated above AIRS L2 does not focus on UT/LS water vapor, though the process of cloud clearing should work best in the non-homogeneous conditions that occur when there are DCC mixed in with overshooting clouds. Perhaps you could introduce the names (and acronyms) of current sounders in line 23, around where you mention Susskind's paper etc.

**Done. The introduction is greatly expanded. See L53-71.**

3. Page 2, line 29 : Please also include Irion, F. W., Kahn, B. H., Schreier, M. M., Fetzer, E. J., Fishbein, E., Fu, D., Kalmus, P., Wilson, R. C., Wong, S., and Yue, Q.: Single-footprint retrievals of temperature, water vapor and cloud properties from AIRS, Atmos. Meas. Tech., 11, 971{995, https://doi.org/10.5194/amt-11-971-2018, 2018 Done. Abstract and introduction are greatly expanded, following this comment. See L53-71.

4. Page 3, line 85-90 : Would you consider showing a MODIS image, and an AIRS BT1231 cm-1 image for this May 16, 2015 storm? I can see AIRS granule 41 for that date would work (see also coincident MODIS 4.05 AM image). This maybe a good check of the realism of the simulations (though yours were performed for about 7 AM of that day). I did a quick check and there were 242 AIRS observations with BT1231 colder than 200 K.

We select this May 16, 2015 storm actually due to the collocation of AIRS granule 41. But collocation between AIRS and CloudSat went bad after 2015 February, therefore, we did not perform further validation tests using actual AIRS observation. We did not use simulation outputs at 4:05 am because the model has to spin up for six hours to properly generate cloud fields. I think adding actual observation can be somewhat misleading, as it is hard to explain why there is a quantitative difference between AIRS and GEM simulation in the first place. Therefore, the MODIS picture is not added. These are not mentioned in the text because there are largely not relevant to the focus we aim to present in this work.

5. Page 6, Line 145-146 and Figure 3 : you may want to point out eg 667 cm-1 and 2300-2360 cm-1 are very high altitude temperature sounding channels, whose weighting functions peak way above the tropopause (and hence the clouds you put in); similarly 1000 cm-1 region is the ozone sounding region, and O3 weighting functions peak in the stratosphere.

2300-2360 cm-1 is not included in the manuscript. Following your comments, the ozone absorption channel is removed as well because ozone is not retrieved. Similar for methane. See L175-180.

radius and IWC jacobians are very similar, so it is quite difficult to unscramble them, other than constraining the effective radius in your retrieval. For completeness, you should discuss what the cloud top/cloud bottom jacobians look like.

We now add a figure showing the normalized spectral uncertainties caused by effective radius and IWC. It is found that signals from effective radii might be distinguishable through a spectral tilting pattern. There was a technical mistake in calculating DFS for effective radius. We now correct it and find that the DFS in effective radius is around 0.6. Please check Section 2.2.1 and Section 2.3.

The TOA spectra are not affected by cloud bottom (in this tropical deep convective system). In case you are interested, in the revised manuscript, we added an additional test to show whether effective radius changes across different vertical levels matter. The answer is no. Only levels around optical depth of 1 (measured from cloud top) matters to the spectral. Therefore, the effective radius we try to retrieve here is vertically constant for a profile. See L193-205, and L220-235.

7. Page 10, Figure 4 panels (a),(c) refer to my earlier comment, namely current sounders have detectors between 640-2700 cm-1, and the noise levels at 640-720 cm-1 are far larger than you use.

See previous response. In the retrieval study, only AIRS channels are used, strictly following the radiometric noise of the instrument, which is not affected by scenes temperature (except a few channels that are already excluded).

8. Page 10, Figure 4(b) is very interesting. Very often AIRS DCC observations at 1231 cm-1 are far colder than NWP tropopause temperatures, so that kink at 80 mb is required. Are there any consequences to the stability of the atmosphere and the tops of overshooting DCC, since your cold profile snaps back to the average stratospheric profile very rapidly? (by about 75 mb!!!).

You might be interested to check the accompanying paper, Feng and Huang 2021., where we show the cold BT anomaly in CO2 absorption channels above storms, from the AIRS L1b product. This cold signature is found to be typical through our retrieval study performed in Feng and Huang 2021. Your question about the stability of the atmosphere caused by temperature perturbation is very interesting. We will look into it.

9. Page 11, Line 225 : Please give a reference for the mapping from one vertical grid to another (eg the remote sensing book by Rodgers 2000, or some of the TES retrieval algorithm papers by eg Worden or Kuwalik. Irion's 2018 AMT paper mentioned above should also have that).

**Added. Do you refer to the least-square interpolation using the Moore-Penrose inverse? We did not add the reference in the original manuscript because we thought it is commonly used and purely mathematical.**

10. page 14, line 315 : replace \board" with \broad"

**Done.**

11. Figure 5, page 15 : I am impressed by the DOFs and ability of your retrieval to get the overall details correct. However as mentioned above the AIRS NeDT is much larger than you use in your OSSE. *Plus the AIRS channels typically has less information content for higher altitudes. So could you run a quick check of how this would impact retrieval performance?*

**See previous responses. DOFs are not biased due to NedT.**

Yes, indeed AIRS contains less information content for high altitudes (DFS between 0.5-0.8 for water vapor, and between 2-4 for temperature, depending on clouds). This is why we add an additional atmospheric constraint in the framework; otherwise, when it comes into the application with real AIRS observation, the retrieval hardly converges. In the accompanying ACP paper, we found that incorporating ERA5 as the additional atmospheric constraint increases the convergence rate to 80%.

Also, DFS in IWC and effective radius are high, which can be somehow expected from the sensitivity test using DARDAR-Cloud. This is because the TOA observed radiance is strongly contributed by cloud emissions near its top. For thinner clouds, this DFS may decrease.

12. Figure 6, page 16 : please check the caption, I think you mean rows 1-5, not columns 1-5; if so there are other sentences in the caption that need fixing.

**Done.**

[13. Could you mention what CO2 and CH4 ppm value you used for your retrieval?

14. You have not explicitly stated which trace gases you retrieved (ozone? CH4?), if any. This goes back to assuming you are using all channels shown in Figures 3,4. Assuming you did use all channels but did not t eg CH4 and O3 and CO, perhaps you could get improved results if you removed the spectral regions that those gases cover?]

Other graces are fixed at tropical mean values and do not affect the OSSE. Following your comments, we have removed absorption channels of O3 and CH4 and clarify our channel selection. Thank you for pointing it out. We have made necessary changes to the accompanying paper as well.

**Response to Quentin-Libois**

This paper is a follow-up of a previous study (Feng and Huang, 2018) that investigated the feasibility to retrieve stratospheric water vapor profiles above thick convective clouds using infrared hyperspectral observations. The present study explores in details the performances of various retrieval strategies (based on the optimal estimation method) and in particular the added value of using ancillary observations (ice cloud properties from the DARDAR synergistic product and atmospheric profiles in the vicinity of the scene) to retrieve water vapor, temperature, and ice water content profiles. Including vertical information on the distribution of ice near the top of the cloud greatly improves the retrievals compared to the slab-cloud hypothesis where an optically thick cloud is located at a single level in the retrieval algorithm, mostly when the top of the cloud exhibits low ice water contents. Using additional atmospheric information also improves the retrieval compared to using climatological a priori values.

The various retrieval strategies are generally well presented and detailed. The paper is well written and the figures appropriate. The manuscript, rather technical, perfectly fits for AMT journal. However the abstract lacks information and the introduction is probably too short to highlight the originality of the work compared to previous studies. A particular critical point is the use of the term "OSSE" to describe something that is probably more a standard retrieval, although tested on atmospheric fields originating from an NWP model. The multiplicity and relative complexity of the different cases described makes it sometimes difficult to follow the optimal estimation framework, but local rephrasing can certainly improve the readability. The results should probably put in a broader context to help the reader figure out whether the improvements for the various retrievals mentioned are significant, and how they compare to existing methods. This paper deserves being published in AMT, but the authors are encouraged to make their best to make the reading as easy as possible. Various suggestions are made below towards this.

**Thank you, Quentin Libois, for your detailed and thoughtful comments. Following your comment, we have greatly expanded the abstract and introduction, added necessary description about the forward model, and expanded the discussion on cloud-induced uncertainties.**

**Specific comments**

1) The abstract is not an abstract so far. An abstract should indeed be a very short summary (very similar to the conclusion actually) of the work. It could be a bit longer, emphasizing for instance that generally sounders reject cloudy pixels (at least in NWP models). It should highlight quantitative results. An abstract should be self-sufficient and provides details : what instruments are used, how much are the performances improved etc. ? The abstract needs rewriting.

**Following this comment, we re-wrote the abstract.**

2) The term "OSSE" is used, in particular in the title, to define the type of experiment performed in this study. Although NWP model outputs are used to test the retrieval algorithms, the present study is not an OSSE, in the sense that no forecasts are performed with the NWP models. In general, an OSSE implies to compare two NWP forecasts, one of those including the assimilation of observations from a supplementary instrument compared to the other. The gain in forecast skill between both forecasts can then be attributed to the additional use of this new instrument. See for instance Arnold and Dey (1986).

**We agree that the term "OSSE" is typically used to describe the weather OSSE that assesses the added value from satellite observations to operational observation systems (e.g., NWP forecast).**

However, many studies (such as Feldman et al., 2013, Smith et al., 2020, Crow et al., 2001) also used this term to describe simulation experiments that derive observational signals from numerical weather forecast model or climate model to evaluate the information content carried by the new instruments, without the assimilation of the observations.

Following your suggestion, we use 'simulation experiment' instead.

3) Some descriptions of the retrieval algorithms are not very clear, in particular when it comes to the details of the optimal estimation method, with the definition of observation vectors, state vectors and covariance errors. In particular Case 5 experiment is not sufficiently clear. It is essential that the combination of the text and equations allows the reader to reproduce the algorithm. The same holds for the definition of DFS relative to each state variable for instance. Likewise, the way atmospheric profiles are sometimes modified for sensitivity studies is not always clear (see examples below).

**To avoid confusion, we now refer to Cases 1-4 as 'retrieval strategies', and avoid referring Case 5 as a retrieval. Necessary clarifications on atmospheric profiles for sensitivity studies are added.**

4) The study deliberately focuses on high thick clouds to probe UTLS atmospheric properties. However the title seems to be more general and does not exclude other types of clouds, including for instance low clouds or thin ice clouds. To make the paper more general it'd be useful to extend the study to other types of clouds. On the contrary, if the focus is only on high thick clouds it has to be more clearly stated. More generally the evaluation of retrieval algorithms on a single atmospheric simulation is probably hard to support, although this does not question the algorithm itself.

A focus on high, thick clouds is now stated in the abstract, as well as the introduction. We do not expand it to cloudy-sky cases in general, considering: 1) a general application using a similar single-footprint OE framework has been discussed in Irian et al., 2018 and DeSouza-Machado et al., 2018, and 2) the way our radiative transfer model is configured is to neglect the contribution from lower atmosphere.

**Technical corrections**

1.4: The combined use of OSSE and retrieval is a bit misleading. Although both approaches rely on the optimal estimation, retrieval aims at retrieving optimal parameters, while OSSE generally results in weather forecasts that can be compared to forecasts without the novel instrumentation included. **See response 2.**

[1.5: "Detecting" is unclear or at least too qualitative, and different than retrieving (quantitative)

1.12: *and* ice particles] Abstract has been re-wrote.

[1.17: what do you include in "atmospheric composition"?

1.22: "severe" is qualitative, what does it mean in terms of cloud fraction or optical thickness for instance? No clear-sky portion in nine (3x3) FOVs (Overcast cloud cover)

1.22: more details should be given on cloud-clearing, and why clouds generally make it difficult to retrieve atmospheric composition. More generally demonstrate more convincingly that what you're proposing has never been done before.

Done

1.24: "large sampling footprint" is unclear. Do you mean vertical resolution? Give some numbers to illustrate that resolution is too coarse, and detail what's the typical vertical resolution that you need to investigate water vapor injection from convective towers

Horizontal resolution. Done.

We re-wrote this paragraph to describe the current in-situ and satellite observations in UTLS water vapor. See L22-71.

1.26: clarify "single footprint". What quantity do you refer to for "retrieval"? If it is water vapor, make it clear because so far it's mixed with "atmospheric composition"

The cloud-clearing algorithm uses adjacent nine footprints while the references given in this paragraph uses only one footprint. See L57-71.

1.29: how can the reader know whether these DFS values are much or not? We re-phrased this sentence. However, whether DFS values are much depends on readers' study of interest, which is the purpose of giving DFS values.

1.34: maybe just say the "actual complexity of clouds" We re-phrased this sentence.

1.37: rephrase this. Suggestion: "Moreover cloud IR emission, that depends on cloud microphysics (or something similar to explain why it's critical to know the details of the clouds), largely contributes to satellite... We re-phrased this sentence.

1.38: what does this lapse rate look like?

We re-phrased this sentence. See L75-79. Deep convective clouds are typically associated with a cold cloud top, due to radiative and dynamical cooling. It is however warmer within the cloud as a result of latent heat release and radiative heating. Therefore, temperature decreases dramatically near the cloud top.

1.46: precise "*in situ* observations" and clarify with respect to the observations used in Feng and Huang (2018) **To answer your question, the observation used in Feng and Huang 2018 is from Anderson et al., 2012,** which is also cited here. Using the in-situ observations carried by aircraft campaigns some limitations in validations of the retrieval. The 'collocation' is never exact. For safety/technical concerns, aircraft samples around a convective system, instead of flying above them. There will always be a certain time/space mismatch.

1.47: more details about OSSE should be provided in the introduction. It's not clear why this cannot be named "sensitivity study based on synthetic observations from NWP model outputs". OSSE generally refers to NWP to estimate the potential impact of a new type of observations on the quality of the forecast. It implies to run a NWP model with data assimilation.

See response 2.

1.56: why studying a tropical cyclone rather than a convective tower?

See L111-115.

One purpose of developing the retrieval approach is to detect stratosphere humidification. Tropical cyclone events can create horizontally extensive cloud cover with overshooting tops, which the retrieval method described in the manuscript can be effectively applied and then contrast the overshooting tops with the 'background' to show up the effect of overshooting. For mid-latitude storms,

We stated the focus on high, thick clouds. Compared to a single convective tower in the mid-latitude, a single tropical cyclone event can produce large high, thick cloud cover for several hundreds of kilometers. This is also relevant to your later question, 'L80', so that a single event produces relatively large variability in temperature, water vapor, and cloud fields.

This event particular event is selected because of the availability of collocated aircraft campaigns (ATTERX) and AIRS overpass, which can help further validation of the retrieval algorithm. However, we chose not to mention it in the paper because such validation studies are not performed here.

1.60: again this really looks like a retrieval rather than an OSSE. **See response 2.**

1.70: how do the 13 hPa relate to the altitude targeted in the paper? **29.1 km.**

1.71: should any temporal spinup or exclusion of outer points of the domain be mentioned here? See L126.

1.76: can you precise whether you handle partial cloud fractions or only cloud fractions of 0 and 1 at each level

The NWP simulates partial cloud fraction and we consider cloud fraction when generating Figure 1(a). However, we find it is not necessary to consider cloud fraction as in this time instances (Figure 1 (a)),

most deep convective clouds have a cloud fraction of 1. See L145 for the definition of 'overshooting deep convective clouds' from a cloud perspective.

1.79: are these liquid clouds accounted for in the radiative transfer simulations? The information should be put here

No. We add clarification in the text; see L171.

1.81: the generates **Done.**

1.80: is this snapshot used in Figure 1 only, or it's the only one used in the present study? What's the limitation of building and evaluating a retrieval algorithm on a single atmospheric situation?

It is the only one used in the present study. For evaluating high, thick clouds only, we think using this snapshot is sufficient, as it provides enough samples and generates different combinations of temperature, water vapors profiles. By estimating the standard deviation for every 10 minutes simulation throughout 24 hours, we find this snapshot is representative of the variability of this event.

As tropical cyclone systems are unique in generating an outflow that affects a large spatial domain, other overshooting convections in mid-latitude storms can be different. We will incorporate mid-latitudes storms to generate a similar simulation study for the application of the method on convective storms other than tropical cyclones and evaluating more detailly on the averaging kernel and sampling footprints.

1.81: what do you mean by "suitable"?See previous response. Descriptions added.

1.83: it is surprising to look at brightness temperature to see the cloud distribution. Looking at condensed water (liquid + ice) paths or cloud fraction would be more standard. Unless you're interested in the vertical distribution here, in which case this should be made clear.

I agree that the water path is typically used to indicate cloud distribution. However, the purpose of showing Figure 1 is to illustrate a hypothetical infrared satellite image that is very commonly used for inferring the distribution of clouds, especially overshooting and deep convective clouds, from satellite observations. Plus, the infrared radiance is not sensitive to total cloud water in the case of deep convective clouds.

1.85: overshooting should be rigorously defined from a cloud perspective

In the original manuscript, we do look into precipitation, cloud fraction, and the vertical extent to identify overshooting from a cloud perspective. It was explicitly stated in L145.

Not all OT-DCCs pass the BT criterion, because the simulated cloud fields near the edge of DCCs can have very low IWC, even though they meet all the criteria described above. The BT criterion is designed to minimize the false positive rate. We chose not to justify the BT criterion here because it is quantitatively examined by the cited previous studies and by an accompanying paper.

1.85: explain the physics behind this brightness temperature difference (BTD), accounting for temperature profile in the UTLS and water vapor and cloud optical properties at these channels. What kind of BTD are expected?

**Done. See L141.**

1.87: "this" criterion is not clear because it was loosely defined. **Done.**

Figure 1: remove "of upwelling radiances". **Done.**

1.103: clarify what are the pressure levels of MODTRAN, and explain how interpolation from GEM to MODTRAN is performed

**Done. Section 2.2 has been expanded. Although we do not perform interpolation in this simulation experiment. Layers from 13.5 hPa to TOA (0.1hPa) are added to radiative transfer model.**

1.104: "cloud information" is unclear. Clearly state "optical properties". Clarify how liquid clouds are treated when present in GEM.

**Section 2.2 has been expanded.**

1.106: what input is taken from GEM for clouds? Ice water content, liquid water content, number concentrations, effective radius? How are clouds specified in MODTRAN? Via optical thickness, single scattering properties? Please clarify how GEM information (vertical profiles) is converted into MOTRAN-required information

**IWC at every vertical level is inputted. Particle size distribution, effective radius, roughness, habit are used to build a look-up table for the extinction coefficients, single-scattering albedo, and asymmetry factors, defined over ice water content. See L159-172.**

1.109: ok for liquid clouds, but the information probably comes too late (see above). Can you then give the range of optical thicknesses obtained for ice clouds to support this choice **Optical thickness of ice is greater than 100 in a typical DCC sample. See L171.**

1.111: reference for AIRS should appear earlier on **Done.**

1.113: cm-1 should not be italic. Holds elsewhere, including figures captions. **Done.**

1.113: what's the spectral resolution of AIRS? Is the resolution of MODTRAN sufficient to simulate AIRS radiances?

Yes. The Full Width at Half Maximum of the AIRS spectral response function is around 0.4 to 1 cm-1. Berk et al., 2017 (Figure 8) validated the line-by-line algorithm of MODTRAN 6 with AIRS spectral response function. It is more convenient to use MODTRAN for calculating cloudy radiances as it includes DISORT. It is also faster than LBLRTM.

1.114: remove "reference level" **Done.**

1.115: does random mean "normal distribution of noise? **Yes. See L183.**

1.115: would you have any reference to support the uncorrelation between spectral channels? In particular, any spectral shift is expected to alter similarly all channels.

Correlation between spectral channels can be a concern; however, we do not find references/data sources that quantitatively document how spectral channels are correlated. Therefore, synthetic noise is assumed to be uncorrelated and the Se error covariance matrix is assumed to be diagonal.

We also tested whether adding cross-channel correlation to 'forward model uncertainty' (i.e., uncertainty due to the slab-cloud assumption). We found it does not make noticeable differences from the results we presented in the manuscript. Therefore, Se is still considered to be diagonal even after adding the forward model uncertainty.

1.119: is optical thickness conserved when effective radius is varied?

No. Optical thickness is not conserved. See response to 'l.106'. The radiative transfer model is driven by IWC at every level and ice optical properties over IWC. The ice optical properties over IWC are strongly dependent on effective radius. See Section 2.2 for an expanded introduction on the radiative transfer model.

1.121: this sentence is unclear **Rephrased.**

1.123: DARDAR product should have been fully defined at the first occurrence

**Done.**

1.124: clarify whether this DARDAR selection has been done by the authors or in the cited paper **Done.**

1.129: does cloud top correspond to the topmost layer of DARDAR? In the legend of Figure 2 it corresponds to 100 hPa. Make sure that this is consistent **Rephrased.**

1.132: how is roughness quantified? If solid columns are chose, is the sensitivity to habit investigated later on, as suggested in the introduction?

1) roughness has a negligible impact on the spectral range of interest, as it mainly affects the scattering angle.

2) Crystal habit indeed has some impacts and can be important. However, we lack the habit fraction dataset, which makes it difficult to assess the impact. In the updated manuscript, we chose to include the radiance uncertainties due to the neglection of habits other than solid columns in the forward model error.

**See Section 2.2.1.**

1.134: does this average conserve the whole mass of ice? Is the average performed over a single profile (to make it vertically homogeneous) or across all profiles to have comparable profiles?

mass of ice is always converged because IWC and effective radius is both reported by DARDAR-Cloud product and we use them independently. Average is performed across all profiles.

**1.135: clarify whether Re is a profile here**

After the revision, Re is not a profile in the retrieval process but it is a profile for generating the synthetic observation. We now add the radiance uncertainties caused by neglecting the vertical variation in Re to the forward model error.

**See Section 2.2.1.**

1.138: all this paragraph is unclear, it's not clear how the profiles are built to compute the mean and standard deviation of brightness temperatures. In particular because the vertical extent of various clouds is different. After revision, we clarify the t,q profiles used, and also add more description of how effective radius affects the radiance spectral in the radiative transfer calculations. Also see previous response that the description of radiative transfer calculation has been expanded.

**See Section 2.2.1.**

1.144: what are the ranges of IWC and effective radius resulting in the observed variability? See revised Figure 2.

1.147: is there a good reason to compute the spectra over the far-infrared, if not used further? This kind of information could be kept for discussion maybe.

**The far-infrared spectra is kept for readers' interests. We kept this spectral range in the plot. To avoid confusion, the spectral range used in the retrieval is marked in Figure 4.**

Note also that for FORUM you can now cite Palchetti et al. (2020).

**Done.**

1.147: cm-1 should not be italic

**Done. For other units as well.**

1.154: why not to conserve IWP in this slab assumption? Unit should be g  $m^{-3}$ . Is this IWC commonly observed? What about the value of Re in this slab?

1) Conserving IWP in the slab is a good idea. Conserving the IWP in such a slab can have IWC around 15 g/m^3 (mean value in the DARDAR-cloud dataset for overshooting DCCs). This value is much higher than reality. We choose 1.5 g/m^3 to match the upper limit we found around 14 km for DCCs. The choice of IWC actually does not matter much in our study of interest, because any value greater than 0.2 g/m^3 for 500 m can lead to saturation (optical depth of 10). 2) Yes.

3) Re is 34 micros

1.157: not clear what these residuals mean. I'd say that they highlight the uncertainties due to the slab-cloud assumption.

**Done.**

1.161: could this tilted bias be mostly related to the penetration depth of radiation, or equivalently to the absorption coefficient of ice (actual emission comes from different effective heights depending on wavenumber)

**Yes, exactly. The dependence of extinction coefficients on wavenumbers determines the penetration depth of radiation at different wavenumbers.**

1.161: is this statement about RMSE and bias general? In the case you'd just have a spectrally flat bias, bias and RMSE would probably be the same, and would both vanish with debiasing.

It is not flat, because altering IWC and Re affects the optical depth and hence thermal emission, which affects the overall brightness temperature. Debiasing does not help to improve the RMSE caused by slabcloud assumption because the STD is large as well.

Figure 3: probably R(Re0, IWC) for the second line **Done.**

1.170: F/x should probably be derivatives **Done.**

1.176: clarify whether K is computed at  $x_0$ , or whether it's computed iteratively and the final one is that on the best estimate

Done. It is iteratively computed. Although updating K (or not), except at the beginning two time steps, does not make much difference to the retrieved values.

1.178: isn't Eqn. 5.16 of Rodgers (2000) more appropriate? **Done.**

1.183: again, some convincing justification of this uncorrelation would be appreciated **Done.**

Figure 4: *at* a spectral resolution **Done.**

1.187-188: probably redundant **Done.**

1.202: what about the *a priori* on IWC and Re?

IWC uses the same *a priori* dataset as temperature and water vapor. Re is based on the DARDAR-Cloud Datasets. There are no correlation between the two in the covariance matrix. See L350-357.

1.203: it would be interesting to see how the final estimates of Re and IWC differ from the DARDAR *a priori* **Results for Re are added to Table 2 and IWC profiles are added to Figure 7.**

1.205: I don't understand the tentative justification for using log(IWC) in the state vector Mainly to avoid getting negative values and also because DARDAR outputs their posteriori uncertainty in log scale. Actually, we also run retrieval tests using IWC without log scale by adding a positive

**constraints on IWC. The final estimates and DFS do not differ much from what you see in this manuscript.**

1.210: not clear why you perturb the profiles in this way. Could you simply propagate the observation error as it is done for the radiances? In this case the retrieval might be optimistic, but the associated error would reflect the IWC measurement uncertainty. It depends whether you're looking at retrieved values or final uncertainties.

There are DFS in IWC and Re so we would like to update them. In practice, using it as the first guess for iwc or as 'y\_{iwc}' gives similar results (if not identical). We choose to formulate the retrieval this way because we want to use the climatology (a priori) in the x0 and Sa and use observations only in y, to distinguish the source of error.

1.212: microns **Done.**

1.229: what's the rationale for this choice of synthetic observation? Does it correspond to a particular satellite? What if cloud conditions have completely changed? How to compute the observation vector for this? Is unity matrix chosen?

1) This choice is arbitrary.

2) It is possible that cloud conditions have completely changed. In practice, such case will lead to poor convergence. When implement the method using real AIRS observation and ERA5 profiles, the convergence rate reached 80%.

3) In practice, we obtain it from ERA5.

4) Do you refer to a unity matrix for 'forward model' ? Yes, if the observation vector and the state vector have the same vertical profiles. In practice, we use a linear-interpolation matrix or a Moore-Penrose matrix, depending on whether observation vector has finer vertical resolution.

1.235: why not using yatm as the *a priori* instead, and increasing *a priori* error? This holds for DARDAR observations as well.

See previous response and the Turner and Blumberg (2018). In this study, because we do not have information in lower-altitudes from infrared radiances, this framework is especially helpful. The state vector in lower-altitudes sticks to the y\_atm, (i.e., reanalysis). We find that use 'y\_{atm}' as the observation constraint, rather than *a priori*, greatly improves the convergence when using real observations. There might be other technics to improve the convergence, but I find this framework best describes the source of uncertainty.

1.237: what variables are you focusing on to assess the performance? All of them or just water vapor? See Table 2. We use all variables to assess the performance, but water vapor is the focus.

1.242: this experiment is surprising, you just expect to converge to DARDAR and atmospheric observations don't you?

Yes. In the revision, we avoid referring to case 5 as a retrieval. It is the posterior estimation of the state vector without infrared radiances. We conduct it so that readers can statistically compare it to Case 4 and infer the improvement from infrared radiances.

1.244: this short paragraph is surprising. What's the use? Why does it appear now and not earlier? For instance, information content has not been defined so far **Deleted.**

1.247: water vapor "perturbations" is too vague. Where? At which temporal frequency? **See revised Section 2.3.**

1.249: why focusing on slab-cloud assumption only here? **Rephrased it.**

1.257: I guess  $t_{cold}$  should have been defined in the previous paragraph as it is used in Fig. 4. **Done.**

1.257: CO2 not italic

**Done.**

1.264: shouldn't the index 'i+1' be removed? **Done.**

1.266: not clear whether the cases remain the same if you now only keep radiance observations. Do you remove other observations only when computing A, but do the calculation at the retrieval point estimated with all observations? To be clarified

Clarified in Section 3. Yes, I only remove other observation only when computing A, because I want to show the DFS from radiance observation alone.

1.269: "As expected" is not obvious because Fig. 4 suggests that the slab-cloud assumption does not really impact the information content

**Yes, you are right. We rephrased it.**

1.275: maybe say "highlights the strong sensitivity" **Done.**

1.277: reference to Fig. 4b is not obvious. Do you mean reference to slab-cloud error? **Yes. Clarified.**

1.278: here the DFS for IWC with only radiances is considered. Does it prove that there would still be an added value when considering direct observations of IWC through DARDAR? Yes, exactly. Although without suppling y\_iwc (obtained from DARDAR or any other cloud observations), retrieval from mid-infrared spectral radiance does not converge well.

1.279: can you elaborate on this high sensitivity to low IWC on a physical basis? **Done. See L141.**

1.281: that DFS for Re is surprisingly low, even though Figs. 3a and 3b suggest it should be less than DFS for IWC

**Thank you for pointing it out. You are right. There was a mistake in calculating kernels. Corrected in the revision.**

1.288: you probably don't need this recap if the cases have been clearly defined before (and ideally defined when needed, that is not too early) **Deleted.**

1.292: "there are some DFS values" is awkwardly **Rephrased.**

1.301: the patterns are not obvious. Do you refer to differences between the center of the cyclone and the points in the South-East corner of the domain, or within the cyclone? In general, Figure 6 is not as informative as the others in the way it is used in the text.

**Yes. Rephrased.**

1.305: check syntax **Done.**

1.305: where does this 13.5% come from? **exp(-2)**, the transmittance through a layer with optical depth of 2.

Figure 7: why not showing truth and retrieval for IWC? Not clear what 7c and 7f show: true or retrieved profiles?

Clarified in Figure 7. Because truth, retrieval, and y\_iwc are visually similar. It is hard to tell which is which. We added them to Figure 7.

1.311: "broad"

**Done.**

1.311: moistening is not so clear in Fig. 7b for Case 1

Yes, because of the smearing effect as explained. You can still see over a higher mixing ratio over a 40 hPa vertical range.

1.323: "better resemblance of Truth" is awkward.

Done.

Improvement from Case 1 to 3 is not obvious in Fig. 6 for water vapor profile Yes, still because of smearing effect. Including y atm increases DFS, leading to narrower vertical averaging kernel, and therefore a narrower peak in Case 4.

Figure 5: RMSE missing in title of 5f. Not clear why  $y_{IWC}$  is biased. Maybe clarify how it is built, as I was expected some random white noise around the truth.

Thank you for pointing it out. We found that the number generated by a Matlab function was not really random. We have overcome this issue.

1.335: This last sentence is not clear **Rephrased.**

1.338: what is the uncertainty range? Clarifying differences between using observations to refine the a priori or to complete radiance observations would help.

We work on the uncertainty range more on an accompanying paper. Please check Figure A2, https://doi.org/10.5194/acp-2021-154

1.340-341: that's not easy to follow. Consider reexplaining how yatm error is set and whether various errors are tested Done.

1.343: "vertical" variability? Done.

1.354: "synergistic". Also check "synergetic" earlier on Done.

1.372: "best resemble" is awkward **Re-phrased.**

1.377: consider removing the citation to an apparently non-existing paper Removed.

Table 1: in caption, 4 or 5 cases? What does 20 refer to in the DFS? Why are the DFS values similar across cases? Why are DFS values shown here and not in Table 2?

Table 2: in caption, 4 or 5 cases?

- 1) Clarified. We now state four 'retrieval' cases, because case 5 is not a retrieval.
- 2) '20' refers to the number of model layers used to calculate DFS in t and q.
- 3) DFS is similar because uncertainties caused by the slab-cloud assumption is small.
- 4) Because it is hard to merge Table 1 and 2 because the limit of page size.